# DEEP LINEAR HAWKES PROCESSES

## ABSTRACT

Marked temporal point processes (MTPPs) are used to model sequences of different types of events with irregular arrival times, with broad applications ranging from healthcare and social networks to finance. We address shortcomings in existing point process models by drawing connections between modern deep state-space models (SSMs) and linear Hawkes processes (LHPs), culminating in an MTPP that we call the *deep linear Hawkes process* (DLHP). The DLHP modifies the linear differential equations in deep SSMs to be stochastic jump differential equations, akin to LHPs. After discretizing, the resulting recurrence can be implemented efficiently using a parallel scan. This brings parallelism and linear scaling to MTPP models. This contrasts with attention-based MTPPs, which scale quadratically, and RNN-based MTPPs, which do not parallelize across the sequence length. We show empirically that DLHPs match or outperform existing models across a broad range of metrics on eight real-world datasets. Our proposed DLHP model is the first instance of the unique architectural capabilities of SSMs being leveraged to construct a new class of MTPP models.

## 1 INTRODUCTION

Marked temporal point processes (MTPPs) are used to model irregular sequences of events in continuous-time, where each event has an associated type, often referred to as a *mark*. MTPPs model the joint distribution of marked event sequences. They have been successfully applied to modeling purchasing patterns in e-commerce (Türkmen et al., 2019; Vassøy et al., 2019; Yang et al., 2018), patient-specific medical events (Hua et al., 2022), disease propagation (Gajardo & Müller, 2023), and many other domains (Williams et al., 2020; Sharma et al., 2018; Wang et al., 2024).

An MTPP is fully characterized by a *marked intensity process* which specifies the expected instantaneous rate of occurrence of events of each mark conditioned on the event history. State-of-the-art methods use neural networks to compute hidden states that summarize the event history, which are then used to compute marked intensities across future values of time. However, many

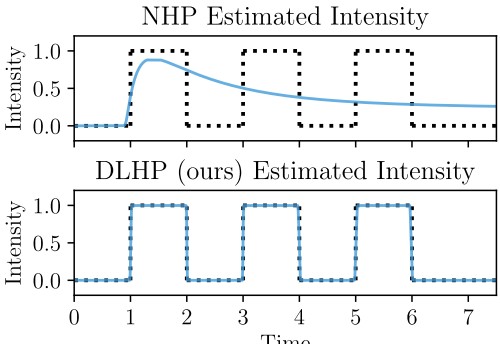

Figure 1: Intensity estimates from trained models when conditioned on an empty sequence $\mathcal{H}_t = \emptyset$ for NHP (Mei & Eisner, 2017) and DLHP, our method. Shown in dotted lines are the ground truth, inhomogeneous Poisson process intensity. Our DLHP is able to accurately capture the background intensity. See Section 5.1 for more details.

models are limited by inexpressive temporal dynamics, lack of support for long-range dependencies, and serial computation (Du et al., 2016; Mei & Eisner, 2017). Recent advances in transformer-based MTPPs have improved performance and gained parallelism, but scale quadratically with sequence lengths (Zhang et al., 2020; Zuo et al., 2020; Yang et al., 2022).

Recently, deep state-space models (often abbreviated as SSMs) have emerged as a challenger to transformer-based models for discrete sequence modeling (Gu et al., 2022b; Smith et al., 2022; Gu & Dao, 2023). SSMs interleave a stack of linear state-space recurrences with position-wise non-linearities (Gu et al., 2021). This architecture has been found to be not only highly performant on a

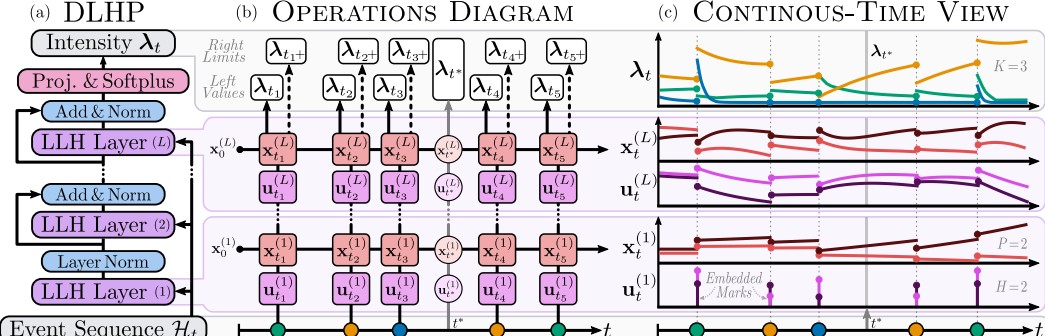

Figure 2: Three different schematics of the *deep linear Hawkes process* (DLHP) and *latent linear Hawkes* (LLH) layer we propose. With increasing granularity: *Left (a)*: On a high level, the DLHP can simply be viewed as a deep stack of neural network layers that transform an event sequence into an intensity function. *Middle (b)*: On a more granular level, individual LLH layers can be viewed as discrete-time recurrences (see Eq. (16)), directly defining an intensity evaluated at select times: $t^*$ using $\mathbf{x}_{t^*}$, right limits $t_i+$ using $\mathbf{x}_{t_i}$, and left values $t_i$ using $\mathbf{x}_{t_i-}$. *Right (c)*: Finally, the same recurrences can be viewed as a set of non-linearly coupled stochastic jump differential equations in continuous-time. Events are embedded and impart impulses to the differential equation. [Added] Note that when decoding intermediate intensities we use a zero-input vector (for both $\mathbf{u}$ input and impulse $\boldsymbol{\alpha}$). We omit the mark-specific impulse for layers 2 to $L$ in this diagram for visual clarity.

wide range of tasks (e.g. Goel et al., 2022; Deng et al., 2024), but retains linear scaling, can be parallelized across the length of a sequence, and can gracefully handle irregularly-spaced observations.

Inspired by this, we revisit a foundational point process model, the linear Hawkes process (LHP Hawkes, 1971), and draw connections between LHPs and deep SSMs. We combine the parameterization and parallelization strategy of SSMs with the functional form of LHPs to create what we call the *deep linear Hawkes process* (DLHP). More formally, the DLHP is a fully-recurrent neural MTPP parameterized by a stack of stochastic jump differential equations on the complex plane (serving as the recurrence) interleaved with position-wise non-linearities (to improve expressivity). This design yields an MTPP with two main advantages over existing neural MTPPs: (i) parallelism across the length of the sequence through the use of parallel scans, and (ii) highly flexible intensity functions. This is achieved not only through the expressivity of the SSM-style architecture, but also by tying the output intensity at time $t$ to the model's continuously-evolved hidden state $\mathbf{x}_t$ (extending ideas from Mei & Eisner (2017) and Yang et al. (2022), see Figs. 1 and 2), and by going beyond the classical LHP form with input-dependent recurrent dynamics (akin to Mamba (Gu & Dao, 2023)).

The contributions of this paper are as follows: We introduce a new family of marked point process models, deep linear Hawkes processes—the first MTPP model that fully leverages the architectural features of deep SSMs. [Edited] We demonstrate that DLHPs match or exceed the performance of existing models across eight real-world datasets, with an average per-event likelihood improvement of **38%** across datasets, over the individually best-performing existing method *on each dataset*. We also verify that DLHP scales more effectively to longer sequences, a crucial capability for a wide range of applications. We release our models, datasets and pipelines as part of the existing `EasyTPP` library (Xue et al., 2023). We conclude by discussing the relative advantages and disadvantages of the DLHP over existing methods, and opportunities for extending this work.

## 2 PRELIMINARIES

### 2.1 MARKED TEMPORAL POINT PROCESSES

Let $t_1, t_2, \cdots \in \mathbb{R}_{\geq 0}$ be a strictly increasing sequence of positive random variables, each representing the time of occurrence for an event of interest.[1] For each $t_i$, let $k_i \in \mathcal{M}$ be a random variable representing accompanying side-information, commonly referred to as an event's *mark*, with $\mathcal{M}$ being

---

[1]Please refer to Tables 2 and 3 in Appendix A for a list of notation and acronym definitions, respectively.

the *mark-space*. In this paper, we focus on discrete and finite mark spaces, i.e. $\mathcal{M} := \{1, \ldots, K\}$; however, in general $\mathcal{M}$ can be continuous or even a mixture of continuous and discrete. Together $t_i$ and $k_i$ fully define a given event. The joint distribution over a sequence of continuous event times and mark types is described as a *marked temporal point process*. We use $\mathcal{H}_t$ to represent the sequence, or *history*, of events up to some time $t$: $\mathcal{H}_t := \{(t_i, k_i) \mid t_i \leq t \text{ for } i \in \mathbb{N}\}$, with $\mathcal{H}_{t-}$ defined similarly except that it does not include events that occur at time $t$.

One way of characterizing an MTPP is through a *marked intensity process*, which describes the instantaneous expected rate of occurrence for events of specific marks. Let $\mathbf{N}_t := [N_t^1, \ldots, N_t^K]^\top \in \mathbb{Z}_{\geq 0}^K$ be the marked counting process which represents the number of occurrences of events of each type of mark in the time span $[0, t]$. The marked intensity process $\boldsymbol{\lambda}_t := [\lambda_t^1, \ldots, \lambda_t^K]^\top \in \mathbb{R}_{\geq 0}^K$ characterizes an MTPP by describing how the counting process changes via:

$$\lambda_t^k \mathrm{d}t := \mathbb{E}\left[\text{event of type } k \text{ occurs in } [t, t + \mathrm{d}t] \mid \mathcal{H}_{t-}\right] = \mathbb{E}\left[N_{t+\mathrm{d}t}^k - N_t^k \mid \mathcal{H}_{t-}\right], \quad (1)$$

with the total intensity $\lambda_t := \sum_{k=1}^K \lambda_t^k$ being the rate that *any* event occurs. Note that the intensity conditions on the left limit of the history $\mathcal{H}_{t-}$ to ensure that the intensity is modeling future events.

Parameterized forms of $\boldsymbol{\lambda}$ are often trained by optimizing the log-likelihood over observed data. The log-likelihood for a single sequence $\mathcal{H}_T$ is defined as (Daley & Vere-Jones, 2003, ch. 7.3):

$$\mathcal{L}(\mathcal{H}_T) := \sum_{i=1}^{|\mathcal{H}_T|} \log \lambda_{t_i}^{k_i} - \int_0^T \lambda_s \mathrm{d}s. \quad (2)$$

**Linear Hawkes Processes**   An (unmarked) *Hawkes* process (Hawkes, 1971), or more generally a *self-exciting* process, is a temporal point process where event occurrences increase the rate at which subsequent events occur soon thereafter. Of particular interest to us are *linear Hawkes processes* (LHPs), which are characterized by the following intensity process:

$$\lambda_t := \nu + \int_{s=0}^{t-} h(t - s) \mathrm{d}N_s := \nu + \sum_{i=1}^{N_{t-}} h(t - t_i), \quad (3)$$

where $\nu > 0$ is the background intensity, $h : \mathbb{R}_{\geq 0} \to \mathbb{R}_{\geq 0}$ is the excitation function (or kernel), and $N_t$ is the associated counting process characterized by intensity $\lambda_t$. $N_{t-}$ is used as the upper limit in Eq. (3) to ensure the intensity at time $t$ does not take into account an event that occurs at time $t$.

Should $h$ correspond to the exponential decay kernel, $h(z) = \alpha \exp(-\beta z)$, then the LHP intensity process is Markov (Law & Viens, 2016) and admits the following stochastic differential form:

$$\mathrm{d}\lambda_t = \beta(\nu - \lambda_{t-})\mathrm{d}t + \alpha \mathrm{d}N_t \iff \lambda_t = \nu + \int_0^{t-} \alpha \exp(-\beta(t - s)) \mathrm{d}N_s \quad (4)$$

$$= \nu + \sum_{i=1}^{N_{t-}} \alpha \exp(-\beta(t - t_i)). \quad (5)$$

LHPs can be extended to the marked setting, with $K$ possible discrete marks, by replacing $\nu$ with a vector of $K$ background rates $\boldsymbol{\nu} := [\nu_1, \ldots, \nu_K]^\top$, and the excitation effect $h(t - s)\mathrm{d}N_s$ with $\mathbf{h}(t - s)\mathrm{d}\mathbf{N}_s$. Here, $\mathbf{h}_{ij}$ of $\mathbf{h} : \mathbb{R}_{\geq 0} \to \mathbb{R}_{\geq 0}^{K \times K}$ describes the excitation that events of type $i$ exerts on future events of type $j$. The counting process, $\mathrm{d}\mathbf{N}_t$, is then either a $K$-dimensional zero-vector if no event occurs at time $t$, or a one-hot vector indicating which mark is associated with the occurring event. Generalizing the exponential kernel to handle marks results in the following differential form:

$$\mathrm{d}\boldsymbol{\lambda}_t = -\boldsymbol{\beta}(\boldsymbol{\lambda}_{t-} - \boldsymbol{\nu})\mathrm{d}t + \boldsymbol{\alpha}\mathrm{d}\mathbf{N}_t, \quad (6)$$

where $\boldsymbol{\beta}, \boldsymbol{\alpha} \in \mathbb{R}_{\geq 0}^{K \times K}$ are restricted to be non-negative to ensure non-negative marked intensities.

## 2.2 STATE-SPACE MODELS

Deep state-space models (SSMs) are a recent innovation in recurrent models that have found success in long-range sequence modeling tasks (Gu et al., 2022b) and language modeling tasks (Gu & Dao, 2023), while also having favorable computational properties. The backbone of deep SSMs is the

linear state-space equations, which define a continuous-time dynamical system with input and output signals $\mathbf{u}(t), \mathbf{y}(t) \in \mathbb{R}^H$, respectively, through linear differential equations:

$$\frac{\mathrm{d}}{\mathrm{d}t}\mathbf{x}(t) = \mathbf{A}\mathbf{x}(t) + \mathbf{B}\mathbf{u}(t) \tag{7}$$

$$\mathbf{y}(t) = \mathbf{C}\mathbf{x}(t) + \mathbf{D}\mathbf{u}(t), \tag{8}$$

where $\mathbf{x}(t) \in \mathbb{R}^P$ is the (hidden) state of the system, and $\mathbf{A} \in \mathbb{R}^{P \times P}, \mathbf{B} \in \mathbb{R}^{P \times H}, \mathbf{C} \in \mathbb{R}^{H \times P}$, and $\mathbf{D} \in \mathbb{R}^{H \times H}$ are the parameters that control the system's dynamics.

Deep SSMs then stack these recurrences interleaved with non-linear position-wise functions, $\sigma$. The function $\sigma$ can contain activation functions, residual connections and normalization layers, and transforms the output $\mathbf{y}$ of the previous recurrence into the input $\mathbf{u}$ of the next, i.e. $\mathbf{u}^{(l)}(t) := \sigma(\mathbf{y}^{(l-1)}(t))$ for layer $l$. This combination yields a sequence model where each recurrence is conditionally linear in time given the input, but is ultimately non-linear in depth due to the function $\sigma$.

To evaluate the SSM, we first discretize the continuous-time system at the desired times, and then evaluate as though it were a conventional discrete-time RNN architecture. Crucially, the linearity of the resulting discrete-time recurrence allows it to be evaluated using parallel scans (Blelloch, 1990; Smith et al., 2022; Gu & Dao, 2023), leading to linear work scaling (i.e. number of operations), and, importantly, sublinear scaling of the computation time with respect to sequence length given sufficient parallel compute. Note this contrasts with conventional sequential RNNs (e.g. LSTMs), which process sequences serially; and attention-based methods, which can be parallelized over a sequence, but have quadratic work scaling with respect to sequence length. This allows SSMs to fully and efficiently leverage modern massively parallel hardware while also a being highly expressive and performant model class. Importantly for our purposes, evaluating a linear recurrence with a parallel scan natively admits evaluations with varying observation intervals. We will leverage this to parsimoniously handle the variable inter-event times observed in MTPP settings.

## 3 DEEP LINEAR HAWKES PROCESSES

In this section, we introduce our *deep linear Hawkes process* (DLHP), a neural MTPP that draws a novel connection between LHPs and deep SSMs. Stochastic jump differential equations, akin to the LHP intensity, form the basis of the conditionally linear recurrent layer, which we refer to as a *latent linear Hawkes* (LLH) layer. The LLH layer can be viewed as a modified SSM recurrence, while still admitting parallel computation. Taking further inspiration from deep SSMs, the DLHP is then made up of a stack of LLH layers, each interleaved with non-linear, position-wise transformations to increase the overall expressivity of the model (see Fig. 2a). In this section, we formalize this approach and outline implementation details.

### 3.1 CONTINUOUS-TIME LATENT LINEAR HAWKES LAYER

We first start by generalizing the intensity of the linear Hawkes process, Eq. (6):

$$\mathrm{d}\boldsymbol{\lambda}_t = -\boldsymbol{\beta}(\boldsymbol{\lambda}_{t-} - \boldsymbol{\nu}_t)\mathrm{d}t + \boldsymbol{\alpha}\mathrm{d}\mathbf{N}_t = -\boldsymbol{\beta}\boldsymbol{\lambda}_{t-}\mathrm{d}t + \boldsymbol{\beta}\boldsymbol{\nu}_t\mathrm{d}t + \boldsymbol{\alpha}\mathrm{d}\mathbf{N}_t, \tag{9}$$

whereby the background intensity $\boldsymbol{\nu}_t$ is allowed to vary over time. If we compare this to the recurrence in Eq. (7), we see that the intensity in the LHP, $\boldsymbol{\lambda}_t$ controlled by decay rates $\boldsymbol{\beta}$, is analogous the state in the linear SSM, $\mathbf{x}(t)$ controlled by state matrix $\mathbf{A}$. Additionally, the time-varying baseline intensity in the LHP, $\boldsymbol{\nu}_t$, is analogous to the SSM input signal, $\mathbf{u}(t)$. What is unique to the LHP is the (mark-specific) impulse $\boldsymbol{\alpha}\mathrm{d}\mathbf{N}_t$. This impulse is important because it allows the model to instantaneously incorporate information from events as they occur, introducing discontinuities in the output signals of the otherwise continuously-integrated system, unlike conventional SSMs.

With this in mind, we adapt Eq. (9) such that it can replace the typical state-space recurrence in an SSM, Eq. (7). To do so, we replace the non-negative $\boldsymbol{\beta}$ with an unrestricted state matrix $\mathbf{A} \in \mathbb{R}^{P \times P}$. Next, given an input signal $\mathbf{u}_t \in \mathbb{R}^H$ we project it to $P$ dimensions with an input matrix $\mathbf{B} \in \mathbb{R}^{P \times H}$ to replace $\boldsymbol{\nu}_t$.[2] What was originally the intensity $\boldsymbol{\lambda}_t$ is now relabeled to be the state of the layer

---

[2]Here, we index time $t$ via subscripts (e.g. $\mathbf{u}_t$) rather than an argument ($\mathbf{u}(t)$) to emphasize that these are stochastic (jump) processes rather than deterministic functions.

$\mathbf{x}_t$. Finally, we allow the impulses to be low-rank by having a shared set of mark embeddings $\boldsymbol{\alpha} \in \mathbb{R}^{R \times K}$ with rank $R$ that are brought into $P$ dimensions with a layer-specific embedding matrix $\mathbf{E} \in \mathbb{R}^{P \times R}$. For simplicity, we set $R = H$ for all our experiments. The equation for the output signal $\mathbf{y}_t$ is left unchanged from Eq. (8), where $\mathbf{C} \in \mathbb{R}^{H \times P}$ and $\mathbf{D} \in \mathbb{R}^{H \times H}$. All of this results in the set of equations that makes up what we call the *latent linear Hawkes* layer:

$$d\mathbf{x}_t = -\mathbf{A}\mathbf{x}_{t-}dt + \mathbf{A}\mathbf{B}\mathbf{u}_{t-}dt + \mathbf{E}\boldsymbol{\alpha}d\mathbf{N}_t \tag{10}$$

$$\mathbf{y}_t = \mathbf{C}\mathbf{x}_t + \mathbf{D}\mathbf{u}_t, \tag{11}$$

where the initial state $\mathbf{x}_0 \in \mathbb{R}^P$ is learned. Realizations of this layer can be seen in Fig. 2c.

## 3.2 CONTINUOUS-TIME DEEP LINEAR HAWKES PROCESS ARCHITECTURE

Inspired by deep SSMs, our MTPP is formed by stacking LLH layers, chaining the output signal $\mathbf{y}$ of one layer to the input $\mathbf{u}$ of another with non-linear transforms in between. The final layer's output is then transformed into the intensity $\boldsymbol{\lambda}$. An illustration of the DLHP architecture is shown in Fig. 2.

Let $L$ be the number of desired LLH layers that comprise a DLHP with input and output signals $\mathbf{u}^{(l)}$ and $\mathbf{y}^{(l)}$ respectively for layers $l = 1, \ldots, L$. For the very first layer, the only input available to condition on are the event occurrences themselves. As such, we set $\mathbf{u}_t^{(1)} = \mathbf{0}$ for all $t \geq 0$.

In general, a layer's output $\mathbf{y}^{(l)} := \text{LLH}^{(l)}(\mathbf{u}^{(l)}, \mathcal{H})$ is passed into a non-linear activation function $f$ (we use $f(z) := \text{GELU}(z)$ (Hendrycks & Gimpel, 2016)), summed with the residual stream $\mathbf{u}^{(l)}$, and normalized with LayerNorm (Ba, 2016) to compute the next layer's input. More formally,

$$\mathbf{u}_t^{(l+1)} := \text{LayerNorm}^{(l)}(f(\mathbf{y}_t^{(l)}) + \mathbf{u}_t^{(l)}) \tag{12}$$

for $t \geq 0$ and $l = 1, \ldots, L$. We use the same strategies for initialization as S5 (Smith et al., 2022), based off the performant HiPPO initialization scheme (Gu et al., 2020). [Added] Due to the transformations, unlike the original LHP, we cannot guarantee the output of the final layer is positive. Therefore, similar to Mei & Eisner (2017), we apply an affine projection followed by a [Added] rectifying transformation to enforce non-negative intensity:

$$\boldsymbol{\lambda}_t := \mathbf{s} \odot \text{softplus}((\mathbf{W}\mathbf{u}_{t-}^{(L+1)} + \mathbf{b}) \odot \mathbf{s}^{-1}) \tag{13}$$

for $t \geq 0$ and where $\mathbf{W} \in \mathbb{R}^{K \times H}$, $\mathbf{b}, \log(\mathbf{s}) \in \mathbb{R}^K$, and $\odot$ is an element-wise product. Eq. (13) implements the "Proj. & Softplus" layer in Fig. 2. The intensity at time $t$ always uses the left-limit of $\mathbf{u}^{(L+1)}$, which in turn uses the left-limit of $\mathbf{y}^{(l)}$ and $\mathbf{u}^{(l)}$ for all $l$ to ensure that it has no information of any events that may or may not have occurred at time $t$ is used.

The DLHP is trained by maximizing the sequence log-likelihood, Eq. (2). Similar to other neural MTPPs, we opt to approximate the integral term in the log-likelihood, $\int_0^T \lambda_s dN_s$, with a Monte-Carlo approximation (Mei & Eisner, 2017). As such, training the model requires the computation of intensity values at event times $t_{1:N}$ and at sampled times $t \sim \mathcal{U}(0, T)$.

[Added] **On the Relationship With the Linear Hawkes Process**   Before discussing how to compute the DLHP and its variations, we briefly reflect on the relationship between the DLHP and LHP. The derivation presented above shows the steps to modify an LHP to be a deep SSM. The connection, parameterization, and equivalence we explore does not materially affect the implementation; this was intended to concretely define how our model differs from a classical model, and to retain the intuition from the simpler LHP (even if the direct interpretability of the LHP is somewhat lost). Alternatively, one could have simply attempted to convert a deep SSM into an MTPP. Likewise, the steps taken and the result would be similar; however, the relationship to other models would be markedly less clear. We hope our exposition makes it clear how the DLHP is a natural extension of a known and well-used model rather than an arbitrary modification to a recent architecture.

## 3.3 DISCRETIZING & DIAGONALIZING THE LLH LAYER

Unlike the LHP intensity, the recurrence in the LLH layer does not permit an analytic solution. As such, we must discretize the continuous-time process to compute values of the layer at specified time points. If we approximate the input signal by treating it as constant over an update interval, also

known as a *zero-order hold* (ZOH) assumption (Iserles, 2009), then we can achieve a closed-form exact update to the recurrence relation. However, unfortunately, this involves a computationally-expensive matrix exponential in the update rule. To circumvent this, we first diagonalize the system and then impose the zero-order hold restriction on it. Doing so converts the matrix exponential into an element-wise exponential operation. This is done for all LLH layers that compose the DLHP. Note that this is same general approach taken by Smith et al. (2022) for deep SSMs.

**Diagonalization**   Let $-\mathbf{A}$ be diagonalizable with a factorization of $\mathbf{V}\mathbf{\Lambda}\mathbf{V}^{-1}$, where $\mathbf{V}, \mathbf{\Lambda} \in \mathbb{C}^{P \times P}$ and $\mathbf{\Lambda}$ is a diagonal matrix of eigenvalues. An equivalent, diagonalized LLH is then

$$d\tilde{\mathbf{x}}_t := \mathbf{\Lambda}\tilde{\mathbf{x}}_{t-}dt + \mathbf{\Lambda}\tilde{\mathbf{B}}\mathbf{u}_{t-}dt + \tilde{\mathbf{E}}\boldsymbol{\alpha}d\mathbf{N}_t \tag{14}$$

$$\mathbf{y}_t := \tilde{\mathbf{C}}\tilde{\mathbf{x}}_t + \mathbf{D}\mathbf{u}_t \tag{15}$$

where $\tilde{\mathbf{x}}_t = \mathbf{V}^{-1}\mathbf{x}_t$, $\tilde{\mathbf{B}} = -\mathbf{V}^{-1}\mathbf{B}$, $\tilde{\mathbf{E}} = \mathbf{V}^{-1}\mathbf{E}$, and $\tilde{\mathbf{C}} = \mathbf{C}\mathbf{V}$. Note that in practice we directly parameterize $\tilde{\mathbf{B}}$, $\tilde{\mathbf{C}}$, and $\tilde{\mathbf{E}}$ to avoid having to learn and invert $\mathbf{V}$. The eigenvalues $\mathbf{\Lambda}$ are also directly parameterized and constrained with negative real-components for stability (Davis, 2013). [Added] While the dynamics are diagonalized, we note this *does not* mean that we are modeling the intensities of different mark types independently. This can be seen two ways: First, the diagonalized dynamics are equivalent to the original dynamics (see Eq. (10), given the system can be diagonalized on the complex plane). Alternatively, the marks interact through the dense input and output matrices, the position-wise non-linearity, the mark embeddings, and the final intensity rectification layer.

**Discretization**   We then employ a ZOH discretization to create a closed-form update from the diagonalized continuous-time system. The ZOH assumption holds the input $\mathbf{u}$ constant over the integration period. This results in the following update rule that transitions from $\mathbf{x}_t$ to $\mathbf{x}_{t'}$, where, by construction, no events occur in $(t, t')$:

$$\tilde{\mathbf{x}}_{t'} := \begin{cases} \bar{\mathbf{\Lambda}}\tilde{\mathbf{x}}_t + (\bar{\mathbf{\Lambda}} - \mathbf{I})\tilde{\mathbf{B}}\mathbf{u}_{t'-} & \text{if no event at } t' \\ \bar{\mathbf{\Lambda}}\tilde{\mathbf{x}}_t + (\bar{\mathbf{\Lambda}} - \mathbf{I})\tilde{\mathbf{B}}\mathbf{u}_{t'-} + \tilde{\mathbf{E}}\boldsymbol{\alpha}_k & \text{if event of type } k \text{ at } t' \end{cases} \tag{16}$$

where $\bar{\mathbf{\Lambda}} := \exp(\mathbf{\Lambda}(t' - t))$ (derivation in Appendix B.2). Please refer to Fig. 2b for an illustration.

Note that the ZOH is an exact update when $\mathbf{u}$ is constant over the window $[t, t']$. While we choose the constant value to set $\mathbf{u}$ to be as $\mathbf{u}_{t'-}$, it is worth noting that technically any value $\mathbf{u}_s$ for $s \in [t, t')$ is valid. We explore this design decision and the impact it has on performance in more detail in Appendices B.4 and D.3. It is important that $\mathbf{u}_{t'}$ is not used as the ZOH value to avoid data leakage.

### 3.4   INPUT-DEPENDENT DYNAMICS

Inspired by recent developments in modern SSMs (e.g. Mamba (Gu & Dao, 2023)), we also consider allowing the dynamics of the system to vary depending on the input and history of previous events. This can allow for more expressive intensities. For instance, dynamically adjusting the real components of $\mathbf{\Lambda}$ to be smaller will result in longer staying power of the recent impulses. Alternatively, larger values will result in more quickly "forgetting" the influence of previous events for a given hidden state channel. This is formalized with the following recurrence relation:

$$d\tilde{\mathbf{x}}_t := \mathbf{\Lambda}_i\tilde{\mathbf{x}}_{t-}dt + \mathbf{\Lambda}_i\tilde{\mathbf{B}}\mathbf{u}_{t-}dt + \tilde{\mathbf{E}}\boldsymbol{\alpha}d\mathbf{N}_t \tag{17}$$

for $t \in (t_i, t_{i+1}]$ where $\mathbf{\Lambda}_i := \text{diag}\left(\text{softplus}(\mathbf{W}'\mathbf{u}_{t_i} + \mathbf{b}')\right)\mathbf{\Lambda}$ with $\mathbf{W}' \in \mathbb{R}^{P \times H}$ and $\mathbf{b}' \in \mathbb{R}^P$. Note that this is still conditionally linear in time as even though $\mathbf{\Lambda}_i$ changes it is entirely input-dependent based on $\mathbf{u}$ and not dependent on previous values of $\mathbf{x}$.

### 3.5   COMPUTING LLH RECURRENCE

Thus far, we have created the LLH layer by diagonalizing and discretizing a modified SSM. As discussed earlier, we would like to take advantage of the efficient parallel scans leveraged by many SSM-based models (Smith et al., 2022; Gu & Dao, 2023; Dao & Gu, 2024). Below we explain how we can still use the parallel scan, despite the modified recurrence.

Parallel scans admit efficient inference over linear recurrences of the form $\mathbf{z}_{i+1} = \mathbf{A}_i\mathbf{z}_i + \mathbf{b}_i$ (Blelloch, 1990). Although we have added an impulse to the recurrence, this is still intrinsically of this

form, where $\mathbf{z}_i := \mathbf{x}_{t_i}$, $\mathbf{A}_i := \exp(\mathbf{\Lambda}_i(t_{i+1} - t_i))$, and $\mathbf{b}_i := (\mathbf{A}_i - \mathbf{I})\tilde{\mathbf{B}}\mathbf{u}_{t_{i+1}-} + \tilde{\mathbf{E}}\boldsymbol{\alpha}_{k_{i+1}}$. As a result, we can leverage efficient parallel scans to compute the sequence of right-limits $\mathbf{x}_{t_{1:N}}$ in parallel across the sequence length. The corresponding left-limits $\mathbf{x}_{t_{1:N}-}$ can then be efficiently computed after by subtracting off $\tilde{\mathbf{E}}\boldsymbol{\alpha}_{k_{1:N}}$ from $\mathbf{x}_{t_{1:N}}$. In Algorithms 1 to 3 we compactly detail how to use a parallel scan to compute the sequence of right limits given events; how to evolve those right limits to compute left limits; and then how to subsequently compute the log-likelihood of the sequence.

## 4 RELATED WORKS

**Neural MTPPs**  Marked temporal point processes (MTPPs) are generative models that jointly model the time and type of continuous-time sequential events, typically characterized by mark-specific intensity functions (Daley & Vere-Jones, 2003). Early approaches, such as self-exciting Hawkes processes (Hawkes, 1971; Liniger, 2009), used simple parametric forms for the intensity. More recently, neural architectures such as RNNs (Du et al., 2016; Mei & Eisner, 2017), CNNs (Zhuzhel et al., 2023), and transformers (Zhang et al., 2020; Zuo et al., 2020; Yang et al., 2022) have been used to more flexibly model the conditional intensity. For intensity-free MTPPs, approaches include normalizing flows (Shchur et al., 2020a; Zagatti et al., 2024), neural processes (Bae et al., 2023), and diffusion models (Zeng et al., 2023; Zhang et al., 2024); however, the most common approach is to model intensities as it requires fewer modeling restrictions.

**Efficient MTPPs**  Due to their recurrent nature, RNN-based MTPP models incur $\mathcal{O}(N)$ complexity for sequences of length $N$ as events must be processed sequentially. Attention-based MTPP models can be applied in parallel across the sequence, but the computational work scales as $\mathcal{O}(N^2)$. Türkmen et al. (2020) proposed modeling events as conditionally independent so long as they occurred within the same time bin of a specified size. This resulted in parallel computation within bins, but still scales overall as $\mathcal{O}(N)$. Shchur et al. (2020b) proposed an intensity-free TPP which uses triangular maps and the time-change theorem (Daley & Vere-Jones, 2003). This was extended by Zagatti et al. (2024) to handle marks, but in doing so, lost many of the benefits of the original model and scales linearly in the mark dimension—which can rapidly become untenable with $O(NK)$ work. To the best of our knowledge, our proposed model is the first that efficiently scales with sequence length and mark space, as well as the first to fully leverage SSMs and parallel scans.

**SSMs for Sequential Modeling**  SSMs have found recent success as alternatives to RNNs, CNNs, and transformers, enjoying reduced training cost and comparable modelling power (Gu et al., 2022b). A range of variants have been developed (Gu et al., 2021; Gupta et al., 2022; Gu et al., 2022a; Smith et al., 2022), and have been applied in language modeling (Gu & Dao, 2023), speech (Goel et al., 2022), and vision (Wang et al., 2023; Zhu et al., 2024). The linear recurrence allows for parallelism, as well as accessible long contexts which would be prohibitive for transformers due to their quadratic scaling. However, SSMs have not previously been applied to MTPPs, in part due to the irregular inter-event times and the input being a stochastic counting process.

[Edited] Concurrent work by Gao et al. (2024) used Mamba (Gu & Dao, 2023), a recent deep SSM architecture, in an MTPP setting, in what they call the *Mamda Hawkes Process* (MHP). The MHP uses a mamba SSM as the encoder in an encoder-decoder architecture, also leveraging the variable interval capabilities. Crucially, however, they use a separate parametric decoder for intermediate intensities (similar to, for instance, the THP). This misses the opportunity to "fully" leverage the SSM architecture, re-using the same variable interval evaluation to evaluate the the intensity.

## 5 EXPERIMENTS

We now evaluate our deep linear Hawkes process model. Our core objectives in using SSMs for MTPP modeling were to define an architecture that is both (a) highly performant in its forecasting ability, and (b) able to leverage efficient parallel compute methods to accelerate inference. To this end, we first present a simple exploration of the ability of different models to represent a periodic intensity function. Then we present the main experiments in this paper, where we evaluate our model against a suite of common MTPP models on a range of datasets of different sizes. We conclude by testing the runtime of our model against a variety of baselines. We find that DLHP systematically outperforms baseline methods both in terms of log-likelihood on held-out test data and runtime across a range of sequence lengths. More results and details are included in Appendices C and D.

Table 1: Per event log-likelihood (↑ is better) results on the held-out test set [added] averaged over five seeds (standard deviations in parentheses); OOM indicates insufficient memory. We **bold** the **best** and underline the runner-up per dataset. We also report the mean rank of models across datasets as a summary metric (↓ is better). DLHP is consistently the best or second best-performing model. Extended results and discussion are presented in Appendix D.1.

| Model | Per-Event Log-Likelihood, $\mathcal{L}_{\text{Total}}$ (nats) | | | | | | | | Avg. Ranking |
| --- | --- | --- | --- | --- | --- | --- | --- | --- | --- |
| | Amazon | Retweet | Taxi | Taobao | StackOverflow | Last.fm | MIMIC-II | EHRShot | |
| RMTPP | -2.136 (0.003) | -7.098 (0.217) | 0.346 (0.002) | 1.003 (0.004) | -2.480 (0.019) | -1.780 (0.005) | -0.472 (0.026) | -8.081 (0.025) | 6.1 |
| NHP | 0.129 (0.012) | **-6.348** (0.000) | 0.514 (0.004) | 1.157 (0.004) | -2.241 (0.002) | -0.574 (0.011) | 0.060 (0.017) | -3.966 (0.058) | 2.9 |
| SAHP | -2.074 (0.029) | -6.708 (0.029) | 0.298 (0.057) | -1.646 (0.083) | -2.341 (0.058) | -1.646 (0.083) | -0.677 (0.072) | -6.804 (0.126) | 5.6 |
| THP | -2.096 (0.002) | -6.659 (0.007) | 0.372 (0.002) | -1.712 (0.011) | -2.338 (0.014) | -1.712 (0.011) | -0.577 (0.011) | -7.208 (0.096) | 5.5 |
| AttNHP | 0.484 (0.077) | -6.499 (0.028) | 0.493 (0.009) | 1.259 (0.022) | -2.194 (0.016) | -0.592 (0.051) | -0.170 (0.077) | OOM | 4.1 |
| IFTPP | 0.496 (0.002) | -10.344 (0.016) | 0.453 (0.002) | **1.318** (0.017) | -2.233 (0.009) | **-0.492** (0.017) | 0.317 (0.052) | -6.596 (0.240) | 2.9 |
| DLHP (Ours) | **0.781** (0.011) | -6.365 (0.003) | **0.522** (0.004) | 1.304 (0.039) | **-2.163** (0.009) | -0.557 (0.046) | **1.243** (0.083) | **-2.512** (0.369) | **1.4** |

**Metrics**  Daley & Vere-Jones (2003, p. 276) state that "testing the model on the basis of its *forecasting performance* amounts to testing the model on the basis of its *likelihood*" (emphasis added). As such, our primary metric of interest to assess model performance is the per-event log-likelihood, $\mathcal{L}_{\text{Total}}$. We also investigate time- and mark-prediction performance through their own log-likelihood values, $\mathcal{L}_{\text{Time}} = \sum_{i=1}^{N} \log \lambda_{t_i} - \int_0^T \lambda_s ds$ and $\mathcal{L}_{\text{Mark}} = \sum_{i=1}^{N} \log(\lambda_{t_i}^{k_i}/\lambda_{t_i})$, respectively, where $\mathcal{L}_{\text{Total}} = \mathcal{L}_{\text{Time}} + \mathcal{L}_{\text{Mark}}$. The log-likelihood of the arrival time characterizes the ability of the model to predict when the next event will arrive. The log-likelihood of the mark is effectively the negative cross-entropy classification loss and measures the ability of the model to predict what types of event will occur given their arrival times. We discuss additional metrics in Appendices D.1 and D.4.

**Models**  We compare our model (DLHP) with six of the most common MTPP models: two RNN-based models (RMTPP (Du et al., 2016), NHP (Mei & Eisner, 2017)), three transformer/attention-based models (THP (Zuo et al., 2020), SAHP (Zhang et al., 2020), AttNHP (Yang et al., 2022)), and one intensity-free model (IFTPP (Shchur et al., 2020a)). In all real-world experiments, extensive grid searches were conducted for hyperparameter tuning with configurations chosen based on validation log-likelihood, [added] with the search spaces designed to roughly control parameter counts across models. Specifics for training, hyperparameters, and architectures are given in Appendix C.

**Libraries and Compute Environment**  We implement our DLHP in the `EasyTPP` library (Xue et al., 2023) and use their implementations of the baseline models. We also use the five standard datasets that `EasyTPP` immediately supports (see Appendix C.2 for more details). We then further include three larger datasets to stress-test the MTPP models (see Section 5.2). Unless otherwise stated, all models were trained using a single NVIDIA A10 GPU with 24GB of onboard memory.

## 5.1 SYNTHETIC EXPERIMENTS

We start by performing a simple investigation into the expressivity of the DLHP intensity function and the ability to capture background intensities. We train our model and baselines on 5,000 sequences over the time period $[0, 7.5]$ drawn from an unmarked, inhomogeneous Poisson process with a square-wave intensity function, $\lambda_t := \mathbb{1}(t \in (1, 2) \cup (3, 4) \cup (5, 6))$ (see Fig. 1). We plot the estimated intensity functions conditioned on no events occurring, i.e. $\mathcal{H}_t := \emptyset \; \forall t$.

Intensity estimates are shown for NHP and DLHP specifically in Fig. 1 (and for all models in Fig. 6 in Appendix D.2). We can see that our model successfully captures the true, underlying background intensity process almost perfectly. This is largely attributed to the expressivity of the linear recurrences and non-linear depth of the model. Other models have various failure modes: struggling to capture the multi-modality (RMTPP, NHP, SAHP, and THP), not matching the square shape (previous four and IFTPP), or not being able to stop the pattern from repeating a fourth time (AttNHP).

[Added] We also examine the performance of our DLHP on randomly instantiated parametric Hawkes processes, finding that DLHP is able to successfully recover the ground truth intensity. Furthermore, in this setting DLHP achieves the best held-out log-likelihood scores and competitive next-event time prediction (see Appendix D.5 for more details). These simple experiments confirm that the DLHP is sufficiently expressive to be able to represent more complicated intensity functions while other methods break down.

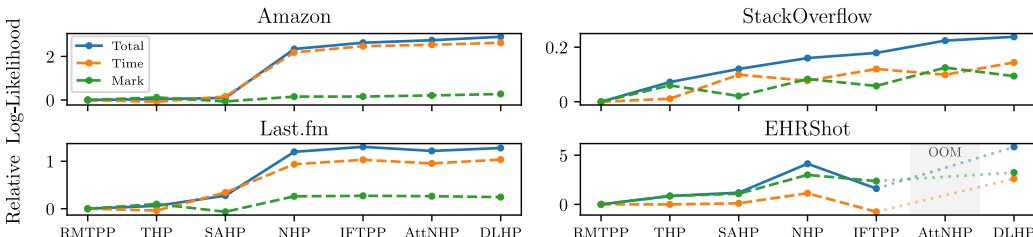

Figure 3: Per event log-likelihood on the held-out test data in Table 1 decomposed into time and mark components (i.e. $\mathcal{L}_{\text{Total}} = \mathcal{L}_{\text{Time}} + \mathcal{L}_{\text{Mark}}$). Models are ordered by their average ranking. Model results are adjusted by subtracting the log-likelihood achieved by RMTPP for readability.

## 5.2 Log-likelihood Results on Real-world Datasets

We empirically investigate the performance of our proposed model against baseline methods by comparing the held-out log-likelihood per event. We evaluate our model on eight real-world datasets. Five of which are taken directly from `EasyTPP` (Xue et al., 2023). We also include two MTPP datasets that have been widely used throughout the literature: **Last.fm**, which includes data on users' music listening patterns from Celma Herrada et al. (2009), and **MIMIC-II**, a subset of de-identified patient hospital visits processed from (Saeed et al., 2002). Finally, we introduce a third dataset from the recently released, publicly available electronic health record (EHR) dataset **EHRShot** (Wornow et al., 2023). To construct the dataset, we first establish the most used *Current Procedural Terminology* (CPT-4) codes that identify medical services and procedures as events. The processed dataset comprises sequences of CPT-4 codes issued to individual patients during their care. This dataset has a maximum sequence length $10\times$ longer than the longest in the `EasyTPP` datasets (and $100\times$ that of MIMIC-II), providing a challenging testbed (in terms of scale) beyond existing datasets. Data statistics and other details including pre-processing are provided in Table 6 and Appendix C.3.

From results shown in Table 1, DLHP consistently achieves the best or the second-best log-likelihood across all datasets. Compared to the best baseline model per-dataset, DLHP produces a (geometric) mean likelihood ratio of 1.4 (corresponding to 40% higher likelihood on true events). We decompose this improvements in Fig. 3, where we see the improvements in log-likelihood are mainly driven by better modeling of time. Extended plots included in Appendix D.1. Given the clear improvement in temporal modeling, we posit that DLHPs are particularly well suited in applications that contain more complex patterns over time. All of these results for DLHP utilize input-dependent dynamics (see Section 3.4). This was found to reliably improve forecasting performance in ablation studies (see Appendix D.3).

[Edited] We also report and discuss additional metrics in the Appendix. We report next-mark classification accuracy and RMSE of next-event arrival time, finding that DLHP matches or outperforms all baselines. We also report model calibration with respect to next event time and mark prediction. Calibration aims to grade the predictive uncertainty of the model (Bosser & Taieb, 2023), which is not captured by other metrics such as mark classification accuracy and time RMSE. On the whole, our model (as well as the baselines) tend to produce well-calibrated time and mark predictions across the datasets. We also include full tables for the likelihood decomposition in Table 7.

## 5.3 Speed Testing

A key motivation for DLHP was to leverage the properties of SSMs to accelerate inference. To test this, we measure the wallclock time for a full forward pass and log-likelihood evaluation on random input sequences with lengths ranging from ten events to one million events. The architectures and mark spaces are the same as the StackOverflow experiments (see Tables 5a and 6). We compare the baselines to our PyTorch `EasyTPP` DLHP implementation, which uses an uncompiled loop, and a standalone JAX DLHP implementation, which uses a parallel scan. Results are shown in Fig. 4. The DLHP is faster than all baseline methods for both forward and log-likelihood evaluation (for all but the shortest sequences). The runtime of NHP always scales linearly. [Edited] THP scales well before reverting to superlinear scaling, and then running out of memory. Interestingly, IFTPP has very fast and fairly constant runtime for short sequences before also running out of memory. We

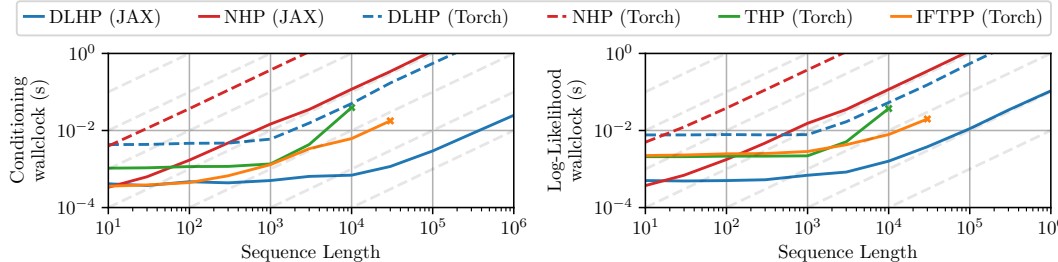

Figure 4: Median runtime, over 10 random seeds, for various models against increasing sequence lengths. We show runtimes for both conditioning on a sequence (Algorithm 1) and likelihood evaluation (Algorithm 3). We see that DLHP is faster across a wide range of sequence lengths.

believe the scaling is due to the highly optimized GRU implementation from PyTorch. As expected, the JAX parallel scan implementation achieves sub-linear scaling in sequence length, and is an order of magnitude faster for conditioning on $N = 10^4$ sequences. Above this, the GPU saturates and reverts to linear scaling. These results confirm that our DLHP can exploit parallel scans to scale to long sequences more effectively than other methods.

## 6 CONCLUSION

We present the *deep linear Hawkes process* (DLHP)—a novel combination of ideas from LHPs and SSMs. Our DLHP leverages the unique properties of deep SSM architectures to achieve a flexible and performant model, without additional and restrictive intensity decoding heads. We then demonstrated that our DLHP outperforms existing methods across a range of standard and new benchmark tasks over various metrics, such as log-likelihood and runtime across sequence lengths. One limitation of our method is the increased complexity of the implementation [Added] (as we require a parallel scan), compared to, for instance, the NHP [Added (A)] (which only requires a basic for loop). Following from this, a second limitation is that we have lost most of the interpretability of the latent dynamics and parameters enjoyed by the LHP. [A] While there is some limited interpretability from the mark-specific impulses due to each LLH layer incrementing the residual stream of the model, this is largely correlative and imprecise. Future research directions therefore include improving on and strengthening these aspects, as well as developing additional theory around the use of deep SSMs in this novel MTPP setting, and [Edited] developing heuristics and best-practices for configuring hyperparameters in various modeling settings. [A] Specifically, exploring the relative benefits of the forward and backward discretization is a unique research direction arising from the DLHP. However, we believe the robustness, performance, computational efficiency, and extensibility of DLHPs make them a very competitive model out-of-the-box for a wide range of applications.

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

# SUPPLEMENTARY MATERIALS FOR SUBMISSION 3981: DEEP LINEAR HAWKES PROCESSES

## TABLE OF CONTENTS

# A ACRONYMS AND NOTATION

Table 2: Key notation used repeatedly across this paper.

| Symbol | Space | Description |
|---|---|---|
| $t$ | $\mathbb{R}_{\geq 0}$ | Time |
| $T$ | $\mathbb{R}_{\geq 0}$ | Maximum time in a given sequence's observation window |
| $t_i$ | $\mathbb{R}_{\geq 0}$ | $i^{\text{th}}$ time |
| $t-$ | $\mathbb{R}_{\geq 0}$ | Subscript minus indicates left-limit |
| $t+$ | $\mathbb{R}_{\geq 0}$ | Subscript plus indicates right-limit |
| $k$ | $\mathcal{M} = \{1, \ldots, K\}$ | Event mark |
| $\mathcal{H}$ | $\mathcal{M}^N \times \mathbb{R}_{\geq 0}^N$ | Event history for $N$ events |
| $\mathbf{N}_t$ | $\mathbb{Z}_{\geq 0}^K$ | Counting process for $K$ marks at time $t$ |
| $\lambda_t^k$ | $\mathbb{R}_{> 0}$ | Intensity of $k^{\text{th}}$ mark type at time $t$ |
| $\boldsymbol{\lambda}_t$ | $\mathbb{R}_{\geq 0}^K$ | Vector of $K$ mark intensities at time $t$ |
| $\lambda_t$ | $\mathbb{R}_{> 0}$ | Ground/total intensity (sum of mark-specific intensities) |
| $\mathcal{L}(\cdot)$ | $\mathbb{R}$ | Log-likelihood of the argument under the model |
| $\nu^{\text{k}}$ | $\mathbb{R}_{> 0}$ | Background intensity for the $k^{\text{th}}$ mark |
| $\boldsymbol{\alpha}$ | $\mathbb{R}_{\geq 0}^{K,K}$ | (For LHP) Matrix of intensity impulses from each type of mark |
| $\boldsymbol{\beta}$ | $\mathbb{R}_{\geq 0}^{K,K}$ | (For LHP) Dynamics matrix of intensity evolution |
| $R$ | $\mathbb{N}$ | Mark embedding rank |
| $P$ | $\mathbb{N}$ | LLH/SSM hidden dimension |
| $\mathbf{x}_t$ | $\mathbb{R}^P$ | LLH/SSM hidden state at time $t$ |
| $\mathbf{x}_0$ | $\mathbb{R}^P$ | Learned LLH/SSM initial hidden state |
| $H$ | $\mathbb{N}$ | LLH/SSM output dimension |
| $\mathbf{y}_t$ | $\mathbb{R}^H$ | LLH/SSM output at time $t$ |
| $\mathbf{u}_t$ | $\mathbb{R}^H$ | LLH/SSM input at time $t$ |
| $\mathbf{A}$ | $\mathbb{R}^{P \times P}$ | LLH/SSM transition matrix |
| $\mathbf{B}$ | $\mathbb{R}^{P \times H}$ | LLH/SSM input matrix |
| $\mathbf{C}$ | $\mathbb{R}^{H \times P}$ | LLH/SSM output matrix |
| $\mathbf{D}$ | $\mathbb{R}^{H \times H}$ | LLH/SSM passthrough matrix |
| $\mathbf{E}$ | $\mathbb{R}^{P \times R}$ | LLH mark embedding matrix ($P \times R$ in low-rank factorization) |
| $L$ | $\mathbb{N}$ | Number of linear recurrences in a DLHP model; model "depth" |
| $\boldsymbol{\alpha}$ | $\mathbb{R}^{R \times K}$ | (For DLHP) Mark impulses ($R \times K$ in low-rank factorization) |
| $\sim$ | N/A | Tilde (e.g. $\tilde{\mathbf{B}}$) denotes variable is in the diagonalized eigenbasis |
| $\Lambda$ | $\mathbb{C}^{P \times P}$ | Matrix of eigenvalues of $\mathbf{A}$; diagonalized dynamics matrix |
| $\bar{\Lambda}$ | $\mathbb{C}^{P \times P}$ | Discretized diagonal dynamics matrix |
| $(l)$ | N/A | Superscript index in parenthesis indicates layer (i.e. $\mathbf{x}$ for layer $l$) |

Table 3: Key acronyms used throughout this paper.

| Acronym | Page number | Definition |
|---|---|---|
| CNN | 6 | Convolutional neural network |
| LHP | 1 | Linear Hawkes process |
| LLH | 2 | Latent linear Hawkes |
| MTPP | 1 | Marked temporal point process |
| RNN | 1 | Recurrent neural network |
| SSM | 1 | (Deep) State-space model |
| TPP | 7 | Temporal point process |
| ZOH | 5 | Zero-order hold |
| RMTPP | 7 | Recurrent marked temporal point process (Du et al., 2016) |
| NHP | 1 | Neural Hawkes process (Mei & Eisner, 2017) |
| SAHP | 7 | Self-attentive Hawkes process (Zhang et al., 2020) |
| THP | 7 | Transformer Hawkes process (Zuo et al., 2020) |
| AttNHP | 7 | Attentive neural Hawkes process (Yang et al., 2022) |
| IFTPP | 7 | Intensity-free temporal point process (Shchur et al., 2020a) |
| DLHP | 1 | Deep linear Hawkes process (ours) |

# B ADDITIONAL DETAILS ON METHODS

## B.1 DLHP ALGORITHMS

---

**Algorithm 1** Deep Linear Hawkes Process: Get Right State Limits

---

**Input:** DLHP layer parameters $\boldsymbol{\Theta} = \left\{ \boldsymbol{\Lambda}^{(l)}, \tilde{\mathbf{B}}^{(l)}, \tilde{\mathbf{C}}^{(l)}, \mathbf{D}^{(l)}, \tilde{\mathbf{E}}^{(l)}, \tilde{\mathbf{x}}_0^{(l)} \right\}_{l=1}^{L}$, event intervals $\Delta t_{1:N}$, nonlinearity $\sigma$, shared mark embeddings $\boldsymbol{\alpha}_{1:N}$.

**Output:** Right state limits $\mathbf{x}_{t_{1:N}}^{(1:L)}$

1: $\mathbf{u}_{t_{1:N}-} = \mathbf{0}$      ▷ Left input limits
2: **for** $l$ **in** $1:L$ **do**
3:     $\bar{\boldsymbol{\Lambda}}_{1:N}^{(l)} = \text{Discretize}\left(\boldsymbol{\Lambda}^{(l)}, \Delta t_{1:N}\right)$      ▷ Zero-order hold, see Eq. (22)
4:     $\tilde{\mathbf{x}}_{t_{1:N}}^{(l)} = \text{ParallelScan}\left(\bar{\boldsymbol{\Lambda}}_{1:N}^{(l)}, (\bar{\boldsymbol{\Lambda}}_{1:N}^{(l)} - \mathbf{I})\tilde{\mathbf{B}}^{(l)}\mathbf{u}_{t_{1:N}-} + \tilde{\mathbf{E}}^{(l)}\boldsymbol{\alpha}_{1:N}\right)$      ▷ Compute right $x$ limits
5:     $\tilde{\mathbf{x}}_{t_{1:N}-}^{(l)} = \tilde{\mathbf{x}}_{t_{1:N}}^{(l)} - \tilde{\mathbf{E}}^{(l)}\boldsymbol{\alpha}_{1:N}$      ▷ Compute left $x$ limits
6:     $\mathbf{u}_{t_{1:N}-} = \text{LayerNorm}\left(\sigma\left(\tilde{\mathbf{C}}^{(l)}\tilde{\mathbf{x}}_{t_{1:N}-} + \mathbf{D}^{(l)}\mathbf{u}_{t_{1:N}-}\right) + \mathbf{u}_{t_{1:N}-}\right)$      ▷ Compute next layer's left $u$ limits
7: **end for**
8: **return** $\mathbf{x}_{t_{1:N}}^{(1:L)}$

---

**Algorithm 2** Deep Linear Hawkes Process: Get Intensity From Right Limit

---

**Input:** DLHP layer parameters $\boldsymbol{\Theta} = \left\{ \boldsymbol{\Lambda}^{(l)}, \tilde{\mathbf{B}}^{(l)}, \tilde{\mathbf{C}}^{(l)}, \mathbf{D}^{(l)}, \tilde{\mathbf{E}}^{(l)}, \tilde{\mathbf{x}}_0^{(l)} \right\}_{l=1}^{L}$, Previous state right limits $\mathbf{x}_t^{(1:L)}$, Integration period $\delta t$, nonlinearity $\sigma$, Intensity function IntensityFn.

**Output:** Intensity left limit $\boldsymbol{\lambda}_{t+\delta t}$

1: $\mathbf{u}_{t+\delta t-} = \mathbf{0}$      ▷ Left input limit
2: **for** $l$ **in** $1:L$ **do**
3:     $\bar{\boldsymbol{\Lambda}}^{(l)} = \text{Discretize}\left(\boldsymbol{\Lambda}^{(l)}, \delta t\right)$      ▷ Zero-order hold, see Eq. (22)
4:     $\tilde{\mathbf{x}}_{t+\delta t-}^{(l)} = \bar{\boldsymbol{\Lambda}}^{(l)}\mathbf{x}_t^{(l)} + (\bar{\boldsymbol{\Lambda}}^{(l)} - \mathbf{I})\tilde{\mathbf{B}}^{(l)}\mathbf{u}_{t+\delta t-}$      ▷ Evolve state
5:     $\mathbf{u}_{t+\delta t-} = \text{LayerNorm}\left(\sigma\left(\tilde{\mathbf{C}}^{(l)}\tilde{\mathbf{x}}_{t+\delta t-}^{(l)} + \mathbf{D}^{(l)}\mathbf{u}_{t+\delta t-}\right) + \mathbf{u}_{t+\delta t-}\right)$      ▷ Compute event left $u$ limits
6: **end for**
7: $\boldsymbol{\lambda}_{t+\delta t} = \text{IntensityFn}(\mathbf{u}_{t+\delta t-})$      ▷ Rectify intensity, see Eq. (13)
8: **return** $\boldsymbol{\lambda}_{t+\delta t}$

---

**Algorithm 3** Deep Linear Hawkes Process: Compute Log-Likelihood

---

**Input:** DLHP layer parameters $\boldsymbol{\Theta} = \left\{ \boldsymbol{\Lambda}^{(l)}, \tilde{\mathbf{B}}^{(l)}, \tilde{\mathbf{C}}^{(l)}, \mathbf{D}^{(l)}, \tilde{\mathbf{E}}^{(l)}, \tilde{\mathbf{x}}_0^{(l)} \right\}_{l=1}^{L}$, Event times $t_{1:N}$, mark types $k_{1:N}$, nonlinearity $\sigma$, shared mark embedding function EmbedMarks, number of integration points per event $M$, Intensity function IntensityFn.

**Output:** Log-ikelihood $\mathcal{L}$

1: $\boldsymbol{\alpha}_{1:N} = \text{EmbedMarks}(k_{1:N})$      ▷ Shared embeddings
2: $t_0 := 0$
3: $\Delta t_{1:N} = t_{1:N} - t_{0:N-1}$
4: $s_{1:N,1:M} \sim \mathcal{U}(0, \Delta t_{1:N})$      ▷ Sample $M$ integration points per interval (non-inclusive)
5: $\tilde{\mathbf{x}}_{t_{1:N}}^{(1:L)} = \text{GetRightStateLimits}(\boldsymbol{\Theta}, \Delta t_{1:N}, \sigma, \boldsymbol{\alpha}_{1:N})$      ▷ Algorithm 1, $\mathcal{O}(\log N)$ parallel time
6: **for** $n$ **in** $1:N$ **do**      ▷ This is *embarrassingly parallelizable* with vmap, $\mathcal{O}(1)$ parallel time
7:     $\boldsymbol{\lambda}_{t_n} = \text{GetIntensityFromRightLimit}\left(\boldsymbol{\Theta}, \tilde{\mathbf{x}}_{t_n}^{(1:L)}, \Delta t_n, \sigma, \text{IntensityFn}\right)$      ▷ Algorithm 2, $\mathcal{O}(1)$ parallel time
8:     **for** $m$ **in** $1:M$ **do**      ▷ This is *embarrassingly parallelizable* with vmap, $\mathcal{O}(1)$ parallel time
9:        $\boldsymbol{\lambda}_{s_{n,m}} = \text{GetIntensityFromRightLimit}\left(\boldsymbol{\Theta}, \tilde{\mathbf{x}}_{t_n}^{(1:L)}, s_{n,m}, \sigma, \text{IntensityFn}\right)$      ▷ Algorithm 2, $\mathcal{O}(1)$ parallel time
10:     **end for**
11: **end for**
12: $\mathcal{L} = \sum_{n=1}^{N} \log \lambda_{t_n}^{k_n} + \sum_{n=1}^{N} \frac{\Delta t_n}{M} \sum_{m=1}^{M} \sum_{k=1}^{K} \lambda_{s_{n,m}}^{k}$      ▷ Eq. (2) with Monte-Carlo approximation of integral
13: **return** $\mathcal{L}$

## B.2 DISCRETIZATION AND ZERO ORDER HOLD

The linear recurrence is defined in continuous-time. This mirrors the (M)TPP setting, where event times are not on a fixed intervals. We use the zero-order hold (ZOH) discretization method, to convert the continuous-time linear recurrence into a sequence of closed-form updates, given the integration times, that can also be efficiently computed. We refer the reader to Iserles (2009) for a comprehensive introduction to the ZOH transform.

The main assumption of the ZOH discretization is that the input signal is held constant over the time period being integrated. Under this assumption, it is possible to solve for the dynamics and input matrices that yield the correct state at the end of the integration period. For the LLH dynamics in Eq. (10), when no events occur in $(t, t')$, this becomes

$$\mathbf{x}_{t'-} = \int_t^{t'} \mathbf{A}\mathbf{x}_t + \mathbf{A}\mathbf{B}\mathbf{u}_t \mathrm{d}t = \overline{\mathbf{A}}\mathbf{x}_t + \overline{\mathbf{A}\mathbf{B}}\mathbf{u}_t \quad \text{assuming} \quad \mathrm{d}\mathbf{u}_t = \mathbf{0} \in [t, t'], \qquad (18)$$

where the resulting discretized matrices are

$$\overline{\mathbf{A}} = e^{\mathbf{A}\Delta t}, \quad \overline{\mathbf{A}\mathbf{B}} = \mathbf{A}^{-1}(e^{\mathbf{A}\Delta t} - \mathbf{I})\mathbf{A}\mathbf{B}, \quad \text{where} \quad \Delta t = t' - t. \qquad (19)$$

The ZOH does not affect the output or passthrough matrices $\mathbf{C}$ and $\mathbf{D}$. To compute the matrices $\overline{\mathbf{A}}$ and $\overline{\mathbf{A}\mathbf{B}}$ however requires computing a matrix exponential and a matrix inverse. However, Smith et al. (2022) avoid this by diagonalizing the system (also avoiding a dense matrix-matrix multiplication in the parallel scan). The diagonalized dynamics and input matrices are denoted $\mathbf{\Lambda}$ (a diagonal matrix) and $\mathbf{\Lambda}\tilde{\mathbf{B}}$ respectively. In this case, Eq. (19) reduces to

$$\overline{\mathbf{A}} = e^{\mathbf{\Lambda}\Delta t}, \qquad (20)$$

$$\overline{\mathbf{A}\mathbf{B}} = \mathbf{\Lambda}^{-1}(e^{\mathbf{\Lambda}\Delta t} - \mathbf{I})\mathbf{\Lambda}\tilde{\mathbf{B}} \qquad (21)$$

$$= (e^{\mathbf{\Lambda}\Delta t} - \mathbf{I})\tilde{\mathbf{B}} \quad \text{(diagonal matrices commute)} \qquad (22)$$

where $e^{\mathbf{\Lambda}\Delta t}$ is trivially computable as the exponential of the leading diagonal of $\mathbf{\Lambda}\Delta t$. These operations are embarrassingly parallelizable across the sequence length and state dimension given the desired evaluation times.

To contextualize, suppose an event occurs at time $t$, Eq. (22) allows us to exactly (under the constant-input assumption) efficiently evaluate the linear recurrence at subsequent times $t'$. We use this extensively in the DLHP to efficiently evaluate the recurrence (and hence the intensity) at the irregularly-spaced event times and times used to compute the integral term.

It should be noted the discretization was done to compute a left-limit $\mathbf{x}_{t'-}$ from a previous right-limit $\mathbf{x}_t$. Should an event not occur at $t'$, then the left- and right-limits agree and $\mathbf{x}_{t'-} = \mathbf{x}_{t'+} = \mathbf{x}_{t'}$. If an event does occur at time $t'$ with mark $k$, then the left-limit $\mathbf{x}_{t'-}$ can be incremented by $\tilde{\mathbf{E}}\boldsymbol{\alpha}_k$ to compute $\mathbf{x}_{t'+} = \mathbf{x}_{t'}$. This increment from left- to right-limit is exact and leverages no discretization assumption.

## B.3 INTERPRETATION FOR INPUT-DEPENDENT DYNAMICS

Consider the input-dependent recurrence for an LLH layer, as defined in Eq. (17):

$$\mathrm{d}\tilde{\mathbf{x}}_t := \mathbf{\Lambda}_i\tilde{\mathbf{x}}_{t-}\mathrm{d}t + \mathbf{\Lambda}_i\tilde{\mathbf{B}}\mathbf{u}_{t-}\mathrm{d}t + \tilde{\mathbf{E}}\boldsymbol{\alpha}\mathrm{d}\mathbf{N}_t \qquad (23)$$

for $t \in (t_i, t_{i+1}]$ where $\mathbf{\Lambda}_i := \mathrm{diag}(\Delta_i)\mathbf{\Lambda}$ with the input-dependent factor defined as $\Delta_i := \mathrm{softplus}(\mathbf{W}'\mathbf{u}_{t_i} + \mathbf{b}') \in \mathbb{R}_{>0}^P$. This factor can be thought of as the input-dependent relative-time scale for the dynamics. To see this, we first note that for vectors $\mathbf{p}, \mathbf{q} \in \mathbb{R}^d$, the following holds true: $\mathrm{diag}(\mathbf{p})\mathbf{q} = \mathbf{p} \odot \mathbf{q} = \mathbf{q} \odot \mathbf{p}$ where $\odot$ is the Hadamard or element-wise product. It then follows that

$$\mathrm{d}\tilde{\mathbf{x}}_t := \mathbf{\Lambda}_i\tilde{\mathbf{x}}_{t-}\mathrm{d}t + \mathbf{\Lambda}_i\tilde{\mathbf{B}}\mathbf{u}_{t-}\mathrm{d}t + \tilde{\mathbf{E}}\boldsymbol{\alpha}\mathrm{d}\mathbf{N}_t \qquad (24)$$

$$= \mathbf{\Lambda}_i(\tilde{\mathbf{x}}_{t-} + \tilde{\mathbf{B}}\mathbf{u}_{t-})\mathrm{d}t + \tilde{\mathbf{E}}\boldsymbol{\alpha}\mathrm{d}\mathbf{N}_t \qquad (25)$$

$$= \mathrm{diag}(\Delta_i)\mathbf{\Lambda}(\tilde{\mathbf{x}}_{t-} + \tilde{\mathbf{B}}\mathbf{u}_{t-})\mathrm{d}t + \tilde{\mathbf{E}}\boldsymbol{\alpha}\mathrm{d}\mathbf{N}_t \qquad (26)$$

$$= [\mathbf{\Lambda}(\tilde{\mathbf{x}}_{t-} + \tilde{\mathbf{B}}\mathbf{u}_{t-})] \odot (\Delta_i\mathrm{d}t) + \tilde{\mathbf{E}}\boldsymbol{\alpha}\mathrm{d}\mathbf{N}_t. \qquad (27)$$

As shown, the positive vector $\Delta_i$ can be thought of as changing the relative time-scale for each channel in the hidden state $\tilde{\mathbf{x}}$. Large values of $\Delta_i$ will act as if time is passing quickly, encouraging the state to converge to the steady-state sooner. Conversely, smaller values will make time pass more slowly causing the model to retain the influence that prior events have on future ones (for that specific channel in $\tilde{\mathbf{x}}$ at least).

### B.4 FORWARDS AND BACKWARDS ZERO ORDER HOLD DISCRETIZATION

In Section 3.3 we highlighted that the ZOH discretization is exact when $\mathbf{u}_t$ is held constant over the integration window. This raises a unique design question for DLHPs: what constant value should $\mathbf{u}_t$ take on when evolving $\mathbf{x}$ from time $t$ to $t'$? For the first layer of the model, the input is zero by construction, so there is no choice to be made—in fact, since $\mathbf{u}$ is constant for the first layer the updates are exact. However, the input is non-zero at deeper layers, and, crucially, varies over the integration period.

We must therefore decide how to select a $\mathbf{u}$ value over the integration period. This should be a value in (or function of) $\{\mathbf{u}_s \mid s \in [t, t')\}$. Note this is because the value at $t'$, $\mathbf{u}_{t'}$, *cannot* be incorporated as this would cause a data leakage in our model; while values prior to $t$ would discard the most recent mark. For this work, we explore two natural choices: (i) the input value at the beginning of the interval, $\mathbf{u}_t$, and (ii) the left-limit at the end of the interval, $\mathbf{u}_{t'-}$. We illustrate the backwards variant in Fig. 2, where in the rightmost panel, we use the $\mathbf{u}_{t^*}$ values at each layer, as opposed to $\mathbf{u}_{t_3}$. We refer to these options as *forwards* and *backwards* ZOH, respectively. All experiments in the main paper utilize backwards ZOH.

It is not obvious *a priori* which one of these modes is more performant. We therefore conducted an ablation experiment in Table 11. We see that there is little difference between the two methods. We also note that models are learned through this discretization, and so this decision does not mean that a model is "incorrectly discretized" one way or the other, but instead they define subtlety different families of models. Theoretical and empirical investigation of the interpretations of this choice is an interesting area of investigation going forwards, extending the ablations we present in Table 11.

### [ADDED] THEORETICAL COMPLEXITY

We include in Table 4 a brief summary of the theoretical complexity of each of the methods we consider. We break these down by the work, memory complexity and theoretical best parallel application time of the forward pass (used when conditioning on a sequence, the left-hand term of Eq. (2)) and evaluating the integral term in Eq. (2) *given that the forward pass has been completed* (as this is either required by the method, and is nearly always evaluated in conjunction with the forward pass). We then state the limiting best-case theoretical parallelism of the two components.

The reasoning behind this is as as follows:

- The forward pass of RMTPP, NHP and IFTPP use non-linear RNNs, and hence incur memory and work that is linear in the sequence length, and cannot be parallelized. However, they re-use the computed hidden states to compute the integral term, and hence, while they incur work and memory that scales in the sequence length and number of events, this work can be perfectly parallelized. This results in a best-case parallelism of $\mathcal{O}(L)$ (dominated by the forward pass).

- SAHP, THP and AttNHP all use self-attention, and hence have a work and memory that scales quadratically in the sequence length, although this work can be parallelized across the sequence length, resulting in logarithmic parallel depth. SAHP and THP re-use embeddings and a parametric decoder, and hence estimating the integral scales like the RNN, and hence the limiting parallelism is still the forward pass. AttNHP is slightly different in that it re-applies the whole independently attention mechanism for each integration point. However, this work is parallelizeable and hence still reduces to a best-case depth of $\mathcal{O}(\log L)$.

- DLHP is an RNN and hence has linear work and memory in the forward pass, but can be parallelized to a best-case depth of $\mathcal{O}(\log L)$ using the parallel scan. We then re-use the states computed in the forward pass for estimating the integral, which, as with the

other RNN methods, is perfectly parallelizable, resulting in a theoretical parallel depth of $\mathcal{O}(\log L)$.

Note that these figures do not account for the number of layers required by each model, which must be evaluated in sequence.

Table 4: Comparison of methods based on memory and compute complexity. We see that our DLHP matches the best performing baseline in all categories. $L$ denotes to the sequence length, and $M$ denotes to the number of Monte Carlo grid points per-event used in evaluating Eq. (2). As IFTPP is an intensity-free method, it does not need to estimate $\int \lambda_t \mathrm{d}t$ as the other methods do.

| Method | Forward Pass | | | Estimating $\int \lambda_t \mathrm{d}t$ | | | Overall |
| | Memory | Work | Theoretical Parallelism | Memory | Work | Theoretical Parallelism | Theoretical Parallelism |
|---|---|---|---|---|---|---|---|
| RMTPP | $\mathcal{O}(L)$ | $\mathcal{O}(L)$ | $\mathcal{O}(L)$ | $\mathcal{O}(LM)$ | $\mathcal{O}(LM)$ | $\mathcal{O}(1)$ | $\mathcal{O}(L)$ |
| NHP | $\mathcal{O}(L)$ | $\mathcal{O}(L)$ | $\mathcal{O}(L)$ | $\mathcal{O}(LM)$ | $\mathcal{O}(LM)$ | $\mathcal{O}(1)$ | $\mathcal{O}(L)$ |
| SAHP | $\mathcal{O}(L^2)$ | $\mathcal{O}(L^2)$ | $\mathcal{O}(\log L)$ | $\mathcal{O}(LM)$ | $\mathcal{O}(LM)$ | $\mathcal{O}(1)$ | $\mathcal{O}(\log L)$ |
| THP | $\mathcal{O}(L^2)$ | $\mathcal{O}(L^2)$ | $\mathcal{O}(\log L)$ | $\mathcal{O}(LM)$ | $\mathcal{O}(LM)$ | $\mathcal{O}(1)$ | $\mathcal{O}(\log L)$ |
| AttNHP | $\mathcal{O}(L^2)$ | $\mathcal{O}(L^2)$ | $\mathcal{O}(\log L)$ | $\mathcal{O}(L^2M)$ | $\mathcal{O}(L^2M)$ | $\mathcal{O}(\log L)$ | $\mathcal{O}(\log L)$ |
| IFTPP | $\mathcal{O}(L)$ | $\mathcal{O}(L)$ | $\mathcal{O}(L)$ | N/A | N/A | N/A | $\mathcal{O}(L)$ |
| DLHP | $\mathcal{O}(L)$ | $\mathcal{O}(L)$ | $\mathcal{O}(\log L)$ | $\mathcal{O}(LM)$ | $\mathcal{O}(LM)$ | $\mathcal{O}(1)$ | $\mathcal{O}(\log L)$ |

## C EXPERIMENTAL CONFIGURATIONS AND DATASETS

### C.1 TRAINING DETAILS & HYPERPARAMETER CONFIGURATIONS

We apply a grid search for all models on all datasets for hyperparameter tuning. We use a default batch size of 256 for training. For models/datasets that require more memory (e.g. large mark space or long sequences), we reduce the batch size and keep them as consistent as possible among all the models on each dataset. We use the Adam stochastic gradient optimizer (Kingma & Ba, 2015), with a learning rate of 0.01 and a linear warm-up schedule over the first 1% iterations, followed by a cosine decay. Initial experiments showed this setting generally worked well across different models and datasets leads to convergence within 300 epochs. We also clip the gradient norm to have a max norm of 1 for training stability. We use Monte-Carlo samples to estimate the integral in log-likelihood, where we use 10 Monte-Carlo points per event during training.

On the five `EasyTPP` benchmark datasets and MIMIC-II that are smaller in their scales, we choose an extended grid based on the architecture reported in the `EasyTPP` paper. Specifically, we search over hidden states size $h = \{16, 32, 64, 128, 256\}$ for RMTPP, $h = \{32, 64, 128\}$ for NHP, and $h = \{16, 32, 64\}$ for IFTPP. For SAHP, THP, and AttNHP, we searched over all combinations of number of $L = \{1, 2, 3\}$, hidden state size $= \{16, 32, 64, 128\}$, and number of heads $= \{1, 2, 4\}$. Finally, for DLHP, we considered combinations for number of layers $= \{1, 2, 3, 4\}$, $p = \{16, 32, 64, 128\}$ and $h = \{16, 32, 64, 256\}$. We fixed the activation function as GeLU (Hendrycks & Gimpel, 2016) and apply post norm with layer norm (Ba, 2016). We fix the dropout as 0.1 for DLHP on the five core benchmark datasets, and add dropout $= \{0, 0.1\}$ to the grid search for the other three datasets. Due to the scale of Last.fm and EHRShot datasets, we perform a smaller search over architectures that roughly match the parameter counts for all models at three levels: 25k, 50k, 200k, and choose the model with the best validation results. AttNHP has expensive memory requirements that tends to have smaller batch sizes than other models. We were unable to train any AttNHP on EHRShot. The final model architectures used are reported in Table 5a and Table 5b. These configurations are also included in the supplementary code we include.

Table 5: Model architectures for the experiments presented in Table 1

(a) Model architectures for the five `EasyTPP` benchmark datasets.

| Model | Amazon | Retweet | Taxi | Taobao | StackOverflow |
|---|---|---|---|---|---|
| RMTPP | $h = 128$ | $h = 16$ | $h = 128$ | $h = 16$ | $h = 256$ |
| NHP | $h = 128$ | $h = 64$ | $h = 128$ | $h = 128$ | $h = 64$ |
| SAHP | $h = 32, l = 2, \text{heads} = 2$ | $h = 32, l = 3, \text{heads} = 4$ | $h = 16, l = 2, \text{heads} = 4$ | $h = 32, l = 1, \text{heads} = 1$ | $h = 64, l = 1, \text{heads} = 1$ |
| THP | $h = 32, l = 2, \text{heads} = 4$ | $h = 16, l = 3, \text{heads} = 4$ | $h = 128, l = 1, \text{heads} = 4$ | $h = 64, l = 1, \text{heads} = 1$ | $h = 16, l = 2, \text{heads} = 4$ |
| AttNHP | $h = 64, t = 16, l = 2, \text{heads} = 4$ | $h = 16, t = 16, l = 2, \text{heads} = 4$ | $h = 16, t = 16, l = 3, \text{heads} = 4$ | $h = 32, t = 16, l = 3, \text{heads} = 4$ | $h = 32, t = 16, l = 2, \text{heads} = 4$ |
| IFTPP | $h = 64$ | $h = 64$ | $h = 32$ | $h = 64$ | $h = 32$ |
| DLHP | $h = 64, p = 128, l = 2$ | $h = 128, p = 128, l = 2$ | $h = 128, p = 16, l = 4$ | $h = 32, p = 16, l = 4$ | $h = 32, p = 32, l = 3$ |

(b) Model architectures for the additional three benchmark datasets.

| Model | Last.fm | MIMIC-II | EHRShot |
|---|---|---|---|
| RMTPP | $h = 256$ | $h = 128$ | $h = 16$ |
| NHP | $h = 112$ | $h = 128$ | $h = 80$ |
| SAHP | $h = 136, l = 2, \text{heads} = 4$ | $h = 64, l = 2, \text{heads} = 4$ | $h = 8, l = 2, \text{heads} = 4$ |
| THP | $h = 48, l = 2, \text{heads} = 4$ | $h = 32, l = 3, \text{heads} = 4$ | $h = 32, l = 2, \text{heads} = 4$ |
| AttNHP | $h = 28, t = 16, l = 2, \text{heads} = 4$ | $h = 64, t = 16, l = 3, \text{heads} = 2$ | OOM |
| IFTPP | $h = 48$ | $h = 256$ | $h = 16$ |
| DLHP | $h = 144, p = 16, l = 2$ | $h = 256, p = 64, l = 2$ | $h = 128, p = 32, l = 2$ |

### C.2 DATASET STATISTICS

We report the statistics of all eight datasets we used in Table 6. We used the `HuggingFace` version of the five `EasyTPP` datasets. For all datasets, we further ensure the MTPP modeling assumptions are satisfied that no more than two events occur at the same time (i.e. inter-arrival time is strictly positive), and event times do not lie on grid points that are effectively discrete-time events. Dataset descriptions and pre-processing details are provided in Appendix C.3.

Table 6: Statistics of the eight datasets we experiment with.

| Dataset | $K$ | Number of Events | | | Sequence Length | | | Number of Sequences | | |
|---|---|---|---|---|---|---|---|---|---|---|
| | | Train | Valid | Test | Min | Max | Mean | Train | Valid | Test |
| Amazon | 16 | 288,377 | 40,995 | 84,048 | 14 | 94 | 44.8 | 6,454 | 922 | 1,851 |
| Retweet | 3 | 2,176,116 | 215,521 | 218,465 | 50 | 264 | 108.8 | 20,000 | 2,000 | 2,000 |
| Taxi | 10 | 51,584 | 7,404 | 14,820 | 36 | 38 | 37.0 | 1,400 | 200 | 400 |
| Taobao | 17 | 73,483 | 11,472 | 28,455 | 28 | 64 | 56.7 | 1,300 | 200 | 500 |
| StackOverflow | 22 | 90,497 | 25,762 | 26,518 | 41 | 101 | 64.8 | 1,401 | 401 | 401 |
| Last.fm | 120 | 1,534,738 | 344,542 | 336,676 | 6 | 501 | 207.2 | 7,488 | 1,604 | 1,604 |
| MIMIC-II | 75 | 9,619 | 1,253 | 1,223 | 2 | 33 | 3.7 | 2600 | 325 | 325 |
| EHRShot | 668 | 759,141 | 165,237 | 170,147 | 5 | 3,955 | 177.0 | 4,329 | 927 | 927 |

## C.3 DATASET PRE-PROCESSING

We use the default train/validation/test splits for `EasyTPP` benchmark datasets. For MIMIC-II, we copy Du et al. (2016) and keep the 325 test sequences in the test split, and further split the 2,935 training sequences into 2,600 for training and 325 for validation. In our pre-processed datasets, Last.fm and EHRShot, we randomly partition into subsets containing 70%, 15%, 15% of all sequences for training/validation/test respectively. We provide a high-level description of all the datasets we used, followed by our pre-processing procedure of Last.fm and EHRShot in more detail. Note that for datasets that contain concurrent events or effectively discrete times, we apply a small amount of jittering to ensure no modeling assumptions are violated in the MTPP framework.

**Amazon** (Ni et al., 2019) contains user product reviews where product categories are considered as marks. **Retweet** (Zhao et al., 2015) predicts the popularity of a retweet cascade, where the event type is decided by if the retweet comes from users with "small", "medium", or "large" influences, measured by number of followers (Mei & Eisner, 2017). **Taxi** data (Whong, 2014; Mei et al., 2019) uses data from the pickups and dropoffs of New York taxi and the marks are defined as the Cartesian product of five discrete locations and two actions (pickup/dropoff). **Taobao** (Xue et al., 2022) describes the viewing patterns of users on an e-commerce site, where item categories are considered as marks. **StackOverflow** contains the badges (defined as marks) awarded to users on a question-answering website. Finally, **MIMIC-II** (Saeed et al., 2002) records different diseases (used as marks) during hospital visits of patients. We add a small amount of noise to the MIMIC-II event times so that events do not lie on a fixed grid. Both StackOverflow and MIMIC-II datasets were first pre-processed by Du et al. (2016).

**Last.fm** Celma Herrada et al. (2009); McFee et al. (2012) records 992 users' music listening habits that has been widely used in MTPP literature (Kumar et al., 2019; Boyd et al., 2020; Bosser & Taieb, 2023). Mark types are defined as the genres of a song, and each event is a play of a particular genre. Each sequence represents the monthly listening behavior of each user, with sequence lengths between 5 and 500. If the song is associated with multiple genres we select a random one of the genres, resulting in a total of 120 different marks.

**EHRShot** Wornow et al. (2023) is a newly proposed large dataset of longitudinal de-identified patient medical records, and has rich information such as hospital visits, procedures, and measurements. We introduce an MTPP dataset derived from EHRShot, where medical services and procedures are treated as marks, as identified by *Current Procedural Terminology* (CPT-4) codes. Each patient defines an event sequence, and we retain only CPT-4 codes with at least 100 occurrences in the dataset. For the $< 1\%$ events of events where there are more than 10 codes at a single timestamp, we retain the top 10 codes with the most frequencies and discard the rest. We then add a small amount of random noise to the event time to ensure they are not overlapping. This process ensures we still satisfy the MTPP framework, and can reasonably instead compute top-10 accuracy for the next mark prediction. Other work has considered extending the MTPP framework to consider simultaneous event occurrence (Chang et al., 2024). Then we standardize each sequence to start and end with start and end of a sequence events. Note that we do not score these events. Event times are normalized to be in hours. We discard sequences that have less than 5 events and a single timestamp. This leads to the final version of our dataset to have 668 marks, and the sequence lengths range from 5 to 3955 events, reflecting patient histories that can span multiple years. We include the

notebook used for compiling the data we use from the original EHRShot data in the supplementary code submission.

# D ADDITIONAL EXPERIMENTAL RESULTS

## D.1 FULL RESULTS ON BENCHMARK DATASETS

We provide the full log-likelihood results and corresponding plots in Table 7 and Fig. 5 respectively, where we decompose the likelihood into time and mark likelihoods. The improvement of our DLHP model is mainly driven by better modeling of time, though we also often obtain best- or second-best predictive performance on marks from the next event prediction accuracy results conditioned on true event time in Table 8. [Added] We also include root mean square error (RMSE) in Table 9 as one of the commonly used metrics in MTPP literature. Intensity-based methods were sampled using the thinning algorithm and averaged over 64 sampled candidate times when memory allows. For our method, we include a simple linear probe on the output hidden vector to directly estimate the next event time, which is substantially faster in wall-clock runtime than sampling even when accounting for fine-tuning the linear probe parameters. For RMSE, IFTPP is the best performer overall; however, across intensity-based methods we find our model to be competitive. We posit this is due to the simplified parameterization of IFTPP lending itself well to capturing the expected time of the next event. In all other predictive metrics, our model ranks the best averaged over all of the datasets.

In aggregate, our model achieves a 1.38 per-event likelihood ratio between itself and the next best method across all datasets (a 38% improvement in likelihood). This is calculated by computing the mean log-likelihood ratio across all datasets and then exponentiating. Doing so is equivalent to taking the geometric mean across likelihood ratios.

Table 7: Complete per-event log-likelihood (higher is better) results on the held-out test for the eight benchmark datasets we consider, [ADDED] averaged over 5 random seeds. In Table 7a we show the full log-likelihood. We then decompose this log-likelihood into the log-likelihood of the event time in Table 7b, and the time-conditional log-likelihood of the mark type in Table 7c. OOM indicates out of memory; [ADDED] standard deviation in brackets. We highlight the best-performing model in bold and underline the second-best. We also report the average rank of models across datasets as a summary metric (lower is better). DLHP is consistently the best or second best-performing model across all datasets.

(a) Full log-likelihood results (equal to the summation of Table 7b and Table 7c). Extended version of Table 1.

| Model | Per-Event Log-Likelihood, $\mathcal{L}_{\text{Total}}$ (nats) | | | | | | | | Avg. Ranking |
|---|---|---|---|---|---|---|---|---|---|
| | Amazon | Retweet | Taxi | Taobao | StackOverflow | Last.fm | MIMIC-II | EHRShot | |
| RMTPP | -2.136 (0.003) | -7.098 (0.217) | 0.346 (0.002) | 1.003 (0.004) | -2.480 (0.019) | -1.780 (0.005) | -0.472 (0.026) | -8.081 (0.025) | 6.1 |
| NHP | 0.129 (0.012) | **-6.348** (0.000) | 0.514 (0.004) | 1.157 (0.004) | -2.241 (0.002) | -0.574 (0.011) | 0.060 (0.017) | -3.966 (0.058) | 2.9 |
| SAHP | -2.074 (0.029) | -6.708 (0.029) | 0.298 (0.057) | -1.646 (0.083) | -2.341 (0.058) | -1.646 (0.083) | -0.677 (0.072) | -6.804 (0.126) | 5.6 |
| THP | -2.096 (0.002) | -6.659 (0.007) | 0.372 (0.002) | -1.712 (0.011) | -2.338 (0.014) | -1.712 (0.011) | -0.577 (0.011) | -7.208 (0.096) | 5.5 |
| AttNHP | 0.484 (0.077) | -6.499 (0.028) | 0.493 (0.009) | 1.259 (0.022) | -2.194 (0.016) | -0.592 (0.051) | -0.170 (0.077) | OOM | 4.1 |
| IFTPP | 0.496 (0.002) | -10.344 (0.016) | 0.453 (0.002) | **1.318** (0.017) | -2.233 (0.009) | **-0.492** (0.017) | 0.317 (0.052) | -6.596 (0.240) | 2.9 |
| DLHP (Ours) | **0.781** (0.011) | -6.365 (0.003) | **0.522** (0.004) | 1.304 (0.039) | **-2.163** (0.009) | -0.557 (0.046) | **1.243** (0.083) | **-2.512** (0.369) | **1.4** |

(b) Per-event log-likelihood of the event times (higher is better).

| Model | Next Event Time Log-Likelihood, $\mathcal{L}_{\text{Time}}$ (nats) | | | | | | | | Avg. Ranking |
|---|---|---|---|---|---|---|---|---|---|
| | Amazon | Retweet | Taxi | Taobao | StackOverflow | Last.fm | MIMIC-II | EHRShot | |
| RMTPP | 0.011 (0.001) | -6.191 (0.083) | 0.622 (0.002) | 2.428 (0.004) | -0.797 (0.005) | 0.256 (0.007) | -0.188 (0.016) | -1.913 (0.025) | 5.6 |
| NHP | 2.116 (0.009) | **-5.584** (0.001) | 0.727 (0.003) | 2.578 (0.006) | -0.699 (0.002) | 1.198 (0.006) | 0.225 (0.016) | -0.821 (0.045) | 3.1 |
| SAHP | 0.115 (0.049) | -5.872 (0.062) | 0.645 (0.044) | 0.489 (0.078) | -0.703 (0.031) | 0.489 (0.078) | -0.244 (0.040) | -1.801 (0.049) | 4.9 |
| THP | -0.068 (0.002) | -5.874 (0.007) | 0.621 (0.002) | 0.220 (0.010) | -0.772 (0.006) | 0.220 (0.010) | -0.271 (0.004) | -1.921 (0.027) | 6.4 |
| AttNHP | 2.416 (0.092) | -5.726 (0.027) | 0.714 (0.010) | 2.654 (0.007) | -0.684 (0.005) | 1.203 (0.015) | 0.031 (0.055) | OOM | 3.3 |
| IFTPP | 2.483 (0.001) | -9.500 (0.011) | **0.735** (0.002) | 2.708 (0.018) | -0.662 (0.007) | **1.277** (0.016) | 0.555 (0.050) | -2.640 (0.115) | 2.9 |
| DLHP (Ours) | **2.652** (0.009) | -5.598 (0.002) | 0.733 (0.003) | **2.719** (0.038) | **-0.641** (0.003) | 1.257 (0.022) | **1.389** (0.053) | **0.382** (0.362) | **1.4** |

(c) Per event log-likelihood of mark type conditioned on the arrival time (higher is better).

| Model | Per-Event Next Mark Log-Likelihood, $\mathcal{L}_{\text{Mark}}$ (nats) | | | | | | | | Avg. Ranking |
|---|---|---|---|---|---|---|---|---|---|
| | Amazon | Retweet | Taxi | Taobao | StackOverflow | Last.fm | MIMIC-II | EHRShot | |
| RMTPP | -2.147 (0.003) | -0.908 (0.141) | -0.276 (0.000) | -1.425 (0.002) | -1.683 (0.015) | -2.035 (0.004) | -0.284 (0.014) | -6.168 (0.025) | 5.9 |
| NHP | -1.987 (0.003) | **-0.764** (0.000) | -0.213 (0.002) | -1.421 (0.004) | -1.542 (0.001) | -1.772 (0.006) | -0.165 (0.002) | -3.144 (0.016) | 2.4 |
| SAHP | -2.189 (0.030) | -0.836 (0.036) | -0.346 (0.024) | -2.136 (0.070) | -1.638 (0.032) | -2.136 (0.070) | -0.433 (0.031) | -5.003 (0.132) | 6.3 |
| THP | -2.028 (0.002) | -0.785 (0.001) | -0.249 (0.001) | -1.932 (0.006) | -1.566 (0.008) | -1.932 (0.006) | -0.306 (0.009) | -5.287 (0.107) | 4.9 |
| AttNHP | -1.933 (0.024) | -0.773 (0.003) | -0.221 (0.002) | -1.395 (0.016) | **-1.510** (0.013) | -1.795 (0.037) | -0.201 (0.025) | OOM | 2.4 |
| IFTPP | -1.988 (0.001) | -0.844 (0.007) | -0.282 (0.001) | **-1.391** (0.005) | -1.571 (0.003) | **-1.769** (0.004) | -0.239 (0.002) | -3.956 (0.192) | 3.8 |
| DLHP (Ours) | **-1.871** (0.002) | -0.767 (0.000) | **-0.211** (0.002) | -1.415 (0.005) | -1.521 (0.008) | -1.814 (0.025) | **-0.145** (0.040) | **-2.893** (0.009) | **1.9** |

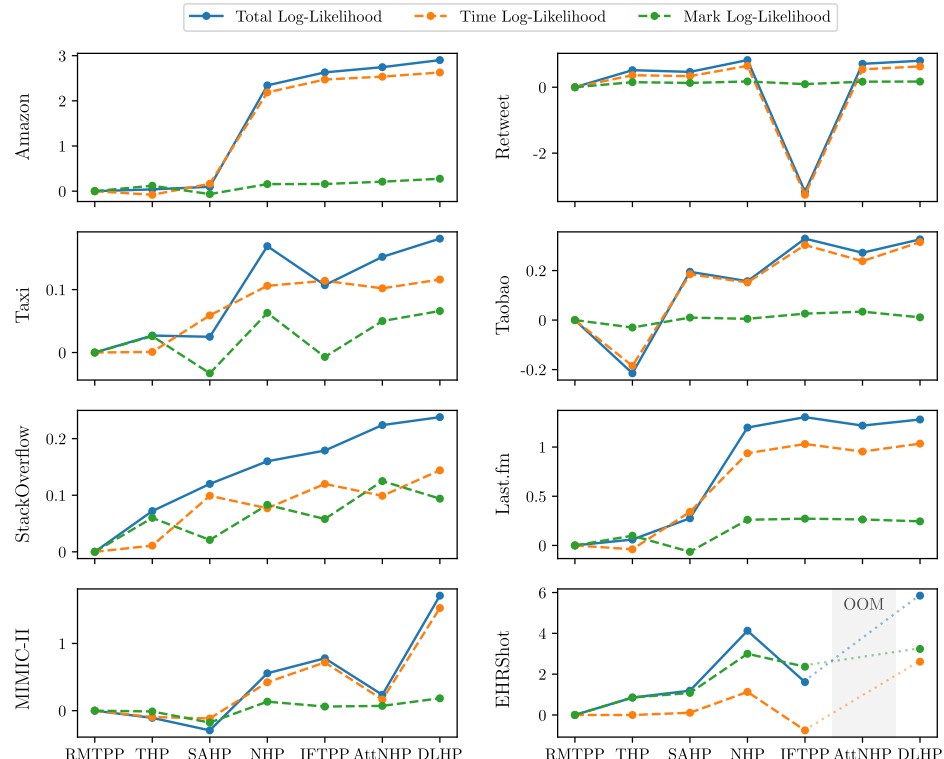

Figure 5: Visualization of $\mathcal{L}_{\text{Total}}$ decomposed into $\mathcal{L}_{\text{Time}}$ and $\mathcal{L}_{\text{Mark}}$ for all models and all datasets relative to RMTPP, as discussed in Section 5.2. The improvement of DLHP is mainly driven by better modeling of $\mathcal{L}_{\text{Time}}$.

Table 8: Next event prediction accuracy (reported as a percentage, ↑ is better) conditioned on the true event time. We report top 1 accuracy for all datasets except for top 10 accuracy for EHRShot, due to the pre-processing procedure described in Appendix C.3. We **bold** the **best** result per dataset, and underline the runner-up.

| Model | Next Mark Accuracy (%) | | | | | | | | Avg. Ranking |
|---|---|---|---|---|---|---|---|---|---|
| | Amazon | Retweet | Taxi | Taobao | StackOverflow | Last.fm | MIMIC-II | EHRShot (Top 10) | |
| RMTPP | 30.96 | 50.36 | 91.37 | 60.93 | 46.46 | 52.51 | 92.20 | 34.09 | 5.63 |
| NHP | 39.23 | **61.47** | 92.82 | **61.58** | 47.03 | 56.43 | 94.32 | 71.85 | 1.88 |
| SAHP | 32.03 | 59.18 | 92.23 | 60.78 | 46.46 | 52.84 | 84.52 | 32.56 | 5.63 |
| THP | 34.63 | 60.17 | 91.59 | 60.00 | 46.64 | 53.28 | 90.98 | 45.47 | 5.13 |
| AttNHP | 38.55 | 60.92 | 92.60 | 61.24 | **48.33** | 56.18 | 91.98 | OOM | 3.00 |
| IFTPP | 35.75 | 49.08 | 91.71 | 60.93 | 45.69 | **56.44** | 93.43 | 60.60 | 4.25 |
| DLHP | **40.66** | 61.33 | **93.05** | 61.06 | 47.45 | 56.26 | **96.55** | **75.45** | **1.75** |

Table 9: Comparison of RMSE of the next-event time prediction on the benchmarks we consider in Section 5.2 in the same format as Table 8. Lower RMSE values indicate better performance.

| Model | Next event time RMSE (↓) | | | | | | | | Avg. Ranking |
|---|---|---|---|---|---|---|---|---|---|
| | Amazon | Retweet | Taxi | Taobao | StackOverflow | Last.fm | MIMIC-II | EHRShot | |
| RMTPP | 0.361 | 16152 | 0.284 | 0.127 | 1.054 | 15.864 | 0.754 | 3445 | 4.8 |
| NHP | 0.342 | 15322 | 0.283 | 0.127 | 1.028 | 15.841 | 0.739 | 3430 | 3.1 |
| SAHP | 0.345 | 16018 | 0.285 | **0.126** | 1.034 | 15.830 | 0.805 | 3418 | 3.8 |
| THP | 0.335 | 15848 | 0.286 | **0.126** | 1.029 | 15.878 | 0.781 | 3504 | 4.3 |
| AttNHP | 1.214 | 16220 | 1.273 | 0.130 | 1.311 | 15.889 | 0.852 | OOM | 7.0 |
| IFTPP | **0.332** | **2568** | **0.280** | **0.126** | **0.975** | **15.550** | **0.713** | **1899** | **1.0** |
| DLHP (Ours, sampling) | 0.395 | 15223 | 0.282 | 0.127 | 1.021 | 15.844 | 0.764 | 3421 | 3.4 |
| DLHP (Ours, linear probe) | 0.337 | 14642 | 0.283 | 0.128 | 1.038 | 15.710 | 0.810 | 3350 | N/A |

## D.2   FULL RESULTS FOR SYNTHETIC POISSON EXPERIMENTS

We present the full results in Fig. 6 for all models regarding the synthetic experiments discussed in Section 5.1. All models are trained until convergence using a set of 5,000 generated sequences, where we use 20 Monte Carlo points per event to estimate the integral of log-likelihood during training to accommodate the sparsity of events. We used small models so they do not overfit; model architecture and parameter counts are reported in Table 10. We plot the background intensity conditioned on empty sequences using 1,000 equidistant grid points between the start and end points. Our model is the only one that perfectly recovers the underlying ground truth intensity, while also using the fewest parameters.

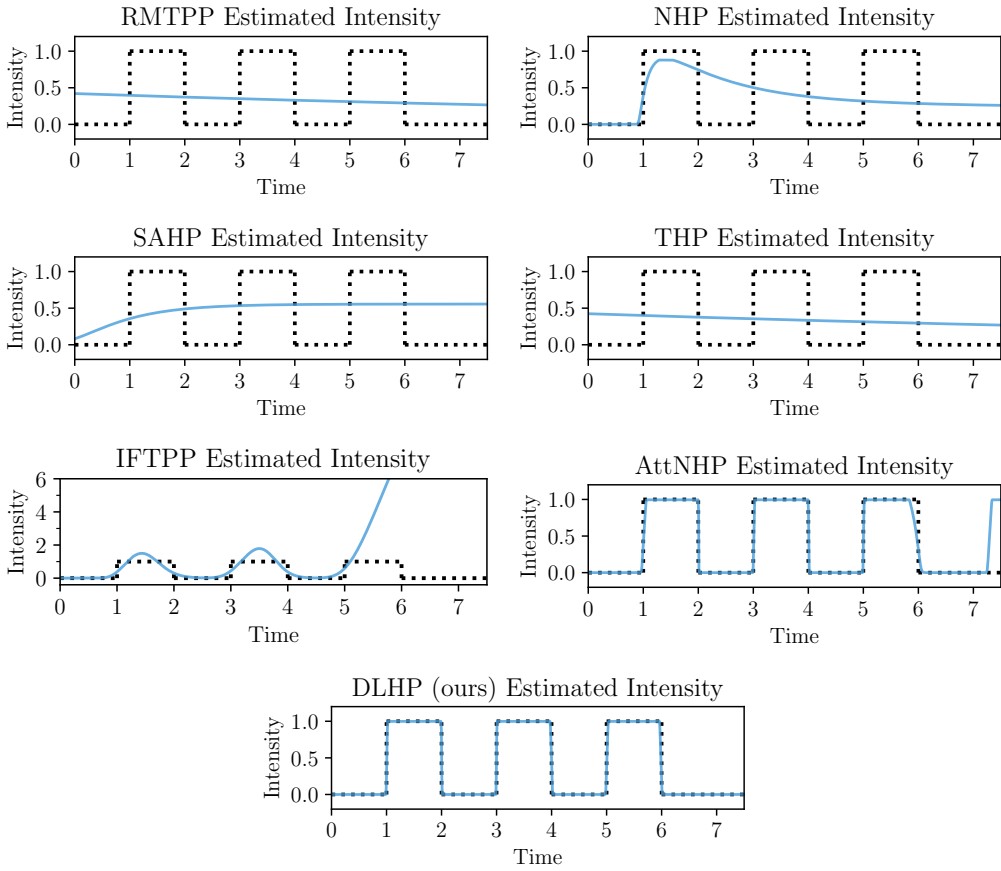

Figure 6: Results for all baseline models for the synthetic Poisson experiment introduced in Section 5.1. The estimated intensity (blue lines) conditioned on an empty sequence are plotted against the ground truth (dotted black lines).

Table 10: Model architectures and corresponding parameter counts for synthetic Poisson experiments.

| Model | Architecture | # Parameters |
|-------|--------------|--------------|
| RMTPP | $h = 16$ | 627 |
| NHP | $h = 8$ | 1010 |
| SAHP | $h = 16, l = 2, \text{heads} = 4$ | 1738 |
| THP | $h = 16, l = 2, \text{heads} = 4$ | 1684 |
| AttNHP | $h = 8, t = 2, l = 2, \text{heads} = 2$ | 1178 |
| IFTPP | $h = 16$ | 1899 |
| DLHP | $h = 4, p = 4, l = 2$ | 178 |

## D.3 ABLATION FOR DIFFERENT DLHP VARIANTS

We perform an ablation study of different model variants that we proposed on all datasets and summarize the results in Table 11. We train EHRShot using 10% of its training data because larger dataset scale requires more training time (but use the original validation and test sets for model selection and reporting results). Forward and backward discretization are very close in performance, with backwards discretization having a slight edge. Models that are input-dependent achieve better performance on most datasets, although on certain datasets input dependence appears to harm performance. It is an interesting direction for future work to explore theoretically and empirically when each of these variants is best. We select backward discretization with input dependence for the results in the main paper.

Table 11: Ablation for different model variants log-likelihood (LL). ID stands for input-dependent, see Section 3.4. Backward and Forward respectively refer to using $\mathbf{u}_{t_{i-1}}$ and $\mathbf{u}_{t_i-}$ (i.e. the previous right limit or current left limit), see Appendix B.4.

| Dataset | Model variant | LL | Arrival time LL | Mark LL conditioned on time |
|---|---|---|---|---|
| Amazon | Forward | 0.705 | 2.617 | -1.912 |
| | Forward + ID | 0.748 | 2.634 | -1.886 |
| | Backward | 0.740 | 2.640 | -1.899 |
| | Backward + ID | **0.765** | 2.638 | -1.873 |
| Retweet | Forward | -6.405 | -5.625 | -0.780 |
| | Forward + ID | -6.370 | -5.602 | -0.767 |
| | Backward | -6.398 | -5.618 | -0.780 |
| | Backward + ID | **-6.367** | -5.600 | -0.767 |
| Taxi | Forward | 0.473 | 0.697 | -0.224 |
| | Forward + ID | 0.525 | 0.733 | -0.208 |
| | Backward | 0.477 | 0.705 | -0.228 |
| | Backward + ID | **0.528** | 0.738 | -0.209 |
| Taobao | Forward | 1.207 | 2.643 | -1.435 |
| | Forward + ID | **1.332** | 2.742 | -1.410 |
| | Backward | 1.215 | 2.648 | -1.432 |
| | Backward + ID | **1.332** | 2.742 | -1.410 |
| StackOverflow | Forward | -2.249 | -0.676 | -1.572 |
| | Forward + ID | -2.174 | -0.644 | -1.530 |
| | Backward | -2.225 | -0.679 | -1.547 |
| | Backward + ID | **-2.165** | -0.636 | -1.529 |
| Last.fm | Forward | **-0.463** | 1.309 | -1.772 |
| | Forward + ID | -0.477 | 1.302 | -1.779 |
| | Backward | -0.474 | 1.303 | -1.777 |
| | Backward + ID | -0.496 | 1.294 | -1.790 |
| MIMIC-II | Forward | 0.555 | 0.847 | -0.292 |
| | Forward + ID | **1.319** | 1.405 | -0.086 |
| | Backward | 0.322 | 0.601 | -0.279 |
| | Backward + ID | 1.231 | 1.345 | -0.114 |
| EHRShot (10%) | Forward | -3.885 | 0.105 | -3.990 |
| | Forward + ID | **-3.848** | -0.021 | -3.827 |
| | Backward | -4.571 | -0.432 | -4.139 |
| | Backward + ID | -4.684 | -0.641 | -4.043 |

### D.4 MODEL CALIBRATION

To further probe the models, we evaluate the calibration metrics of MTPPs that are proposed in literature (Bosser & Taieb, 2023), which has a different focus than log-likelihood-based evaluation. On a high level, calibration describes how well the uncertainty in the model is reflected in the observed data. However, a model can achieve perfect calibration by predicting the marginal distribution, so better calibration *does not* necessarily transform into better predictive performance. We therefore present these metrics as a secondary metric (secondary to log-likelihood per Daley & Vere-Jones (2003)) for investigating the performance of different models. We provide summarized statistics for both probabilistic calibration error (PCE) for time calibration and expected calibration error (ECE) for mark calibration in Table 12, and visualize the calibration curves in Figs. 7 and 8. From our results, all MTPP models are well-calibrated on most of the datasets, especially on mark predictions.

Table 12: Calibration results for the models and datasets tests.

(a) Probabilistic calibration error (PCE) for time calibration in percentage.

| Model | Probabilistic Calibration Error (PCE) | | | | | | | |
|---|---|---|---|---|---|---|---|---|
| | Amazon | Retweet | Taxi | Taobao | StackOverflow | Last.fm | MIMIC-II | EHRShot |
| RMTPP | 13.70 | 4.20 | 3.55 | 10.18 | 1.91 | 11.55 | 3.85 | 13.31 |
| NHP | 7.57 | 0.15 | 0.27 | 7.38 | 1.77 | 4.77 | 6.05 | 8.22 |
| SAHP | 10.86 | 9.75 | 1.73 | 2.88 | 1.14 | 10.89 | 2.79 | 15.05 |
| THP | 12.28 | 5.71 | 3.32 | 16.32 | 2.10 | 10.90 | 1.21 | 14.55 |
| AttNHP | 6.20 | 1.26 | 0.96 | 3.17 | 1.52 | 1.57 | 4.66 | OOM |
| IFTPP | 1.74 | 23.93 | 0.44 | 0.61 | 0.50 | 0.30 | 2.19 | 17.66 |
| DLHP | 3.47 | 0.40 | 0.13 | 2.05 | 0.60 | 1.18 | 8.94 | 12.47 |

(b) Expected calibration error (ECE) for mark calibration in percentage.

| Model | Expected Calibration Error (ECE) | | | | | | | |
|---|---|---|---|---|---|---|---|---|
| | Amazon | Retweet | Taxi | Taobao | StackOverflow | Last.fm | MIMIC-II | EHRShot |
| RMTPP | 6.41 | 5.89 | 2.62 | 1.60 | 1.36 | 2.44 | 1.97 | 9.22 |
| NHP | 6.75 | 0.33 | 0.81 | 4.40 | 1.02 | 4.10 | 1.92 | 2.84 |
| SAHP | 8.36 | 4.74 | 6.96 | 3.00 | 1.12 | 8.55 | 5.77 | 11.09 |
| THP | 2.02 | 1.20 | 1.74 | 6.48 | 0.77 | 2.67 | 1.81 | 11.42 |
| AttNHP | 2.88 | 0.39 | 0.44 | 2.52 | 1.21 | 0.50 | 2.79 | OOM |
| IFTPP | 0.37 | 0.58 | 0.41 | 1.49 | 1.48 | 0.59 | 1.40 | 2.01 |
| DLHP | 1.00 | 0.72 | 0.46 | 1.66 | 2.01 | 0.74 | 2.34 | 1.19 |

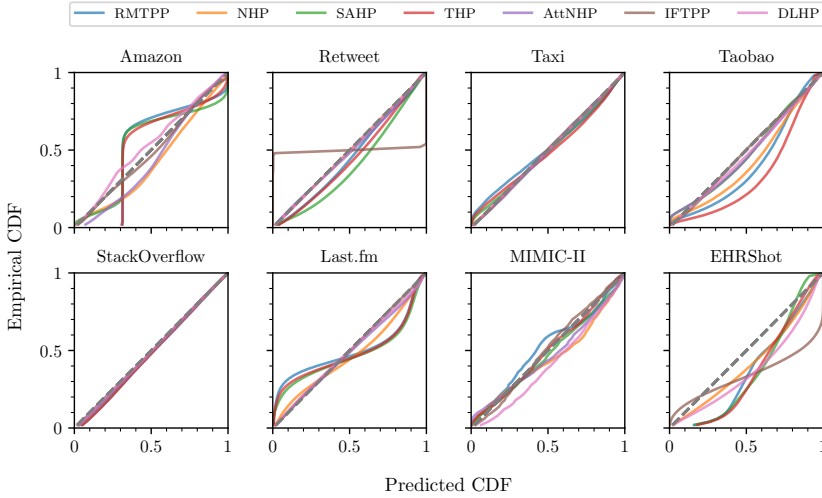

Figure 7: Reliability diagram for predicted inter-arrival time for each model on all datasets. Diagonal dashed lines refer to perfect calibration.

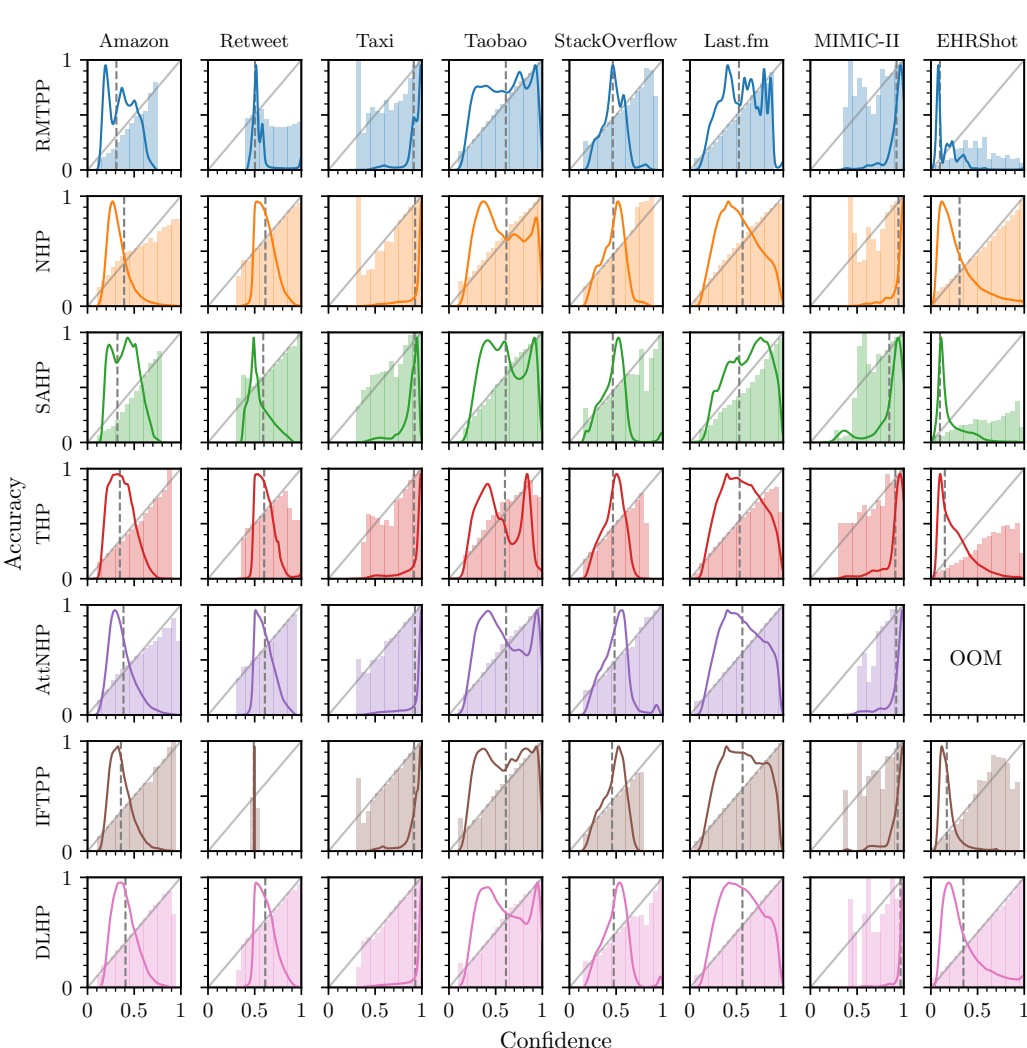

Figure 8: Reliability diagram for mark prediction of all models and all datasets. The $x$-axis specifies the confidence of model estimates grouped into 20 bins, and the $y$-axis of the bar plot is the model accuracy within that bin. The diagonal lines represent perfect calibration. The solid curves depict the distribution of confidences, and do not share the $y$-axis. The grey dashed lines indicate the overall prediction accuracy of the model for the next event conditioned on true event time.

Finally, in Figs. 9 and 10 we plot the log-likelihood of time and mark respectively, versus their corresponding calibration results, to provide an overall view of the performances of different models. Our DLHP model consistently achieves higher log-likelihood while maintaining good calibration on both time and mark components on most datasets.

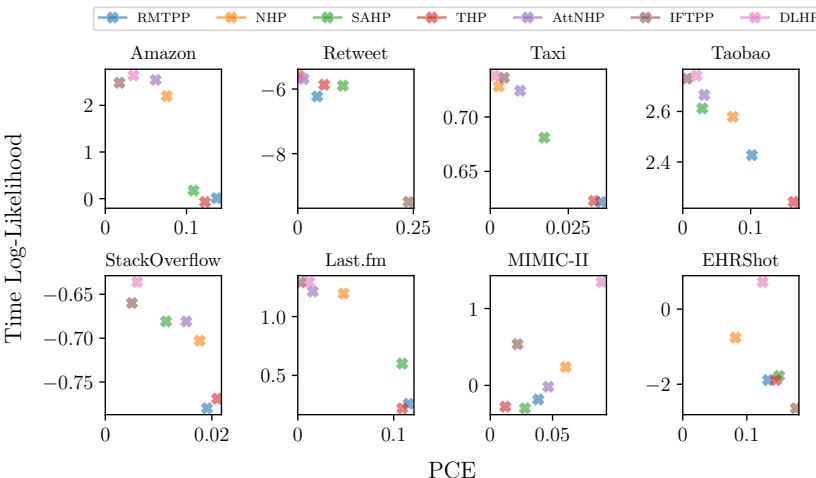

Figure 9: Log-likelihood of time vs. PCE for all models grouped by datasets. Higher log-likelihood and lower PCE are better (i.e. top left corner).

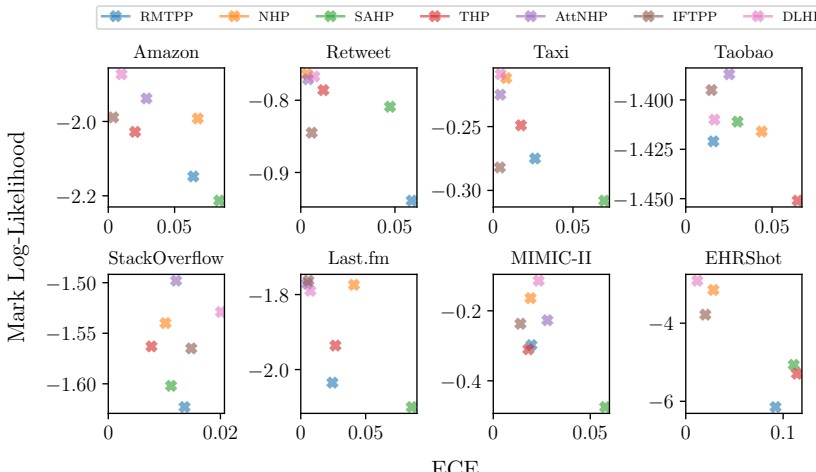

Figure 10: Log-likelihood of mark vs. ECE for all models grouped by datasets. Higher log-likelihood and lower ECE are better (i.e. top left corner).

### D.5 [ADDED] ADDITIONAL SYNTHETIC RESULTS ON MULTIVARIATE HAWKES PROCESSES

We evaluate our model and baseline models against the true model on a randomly initiated parametric Hawkes process with three possible marks. Following the notation in Section 2.1, we draw all parameters from the following distributions: $\boldsymbol{\nu}_i \overset{iid}{\sim} \text{Unif}[0.1, 0.5]$, $\boldsymbol{\alpha}_{ij} \overset{iid}{\sim} \text{Unif}[0.5, 0.8]$, and $\boldsymbol{\beta}_{ij} \overset{iid}{\sim} \text{Unif}[0.4, 1.2]$ for $i, j \in \{1, 2, 3\}$.

All models are trained until convergence using a set of 50,000 generated sequences, where we use 20 Monte Carlo points per event to estimate the integral of log-likelihood during training. Model architecture and parameter counts are reported in Table 13. We plot three example sequences drawn for an additional test set for each model in Figs. 11 and 12, using 1,000 equidistant grid points for any inter-event interval. Dotted lines refer to the intensities under the true underlying parametric model; solid lines are different model estimates from trained models.

As we see in inhomogeneous Poisson processes, our model can recover the ground truth intensities with the fewest parameters. Both neural Hawkes processes and our DLHP show almost perfect recovery of parametric Hawkes processes, especially before seeing any event happening, and at event times. It is also worth noting that our model is 7-9× quicker than NHP and AttNHP regarding wall-clock runtime on a single A5000 GPU. Our results on synthetic experiments validate the model's ability to recover the ground truth intensities.

We further evaluate all models quantitatively using 1,000 test sequences generated from the same multivariate Hawkes process and evaluated both log-likelihood and RMSE for the immediate next event. We see our method competitive again on both metrics.

Table 13: Model architectures and corresponding parameter counts for parametric Hawkes processes experiments.

| Model | Architecture | # Parameters |
|-------|--------------|--------------|
| RMTPP | $h = 16$ | 697 |
| NHP | $h = 8$ | 1046 |
| SAHP | $h = 16, l = 2, \text{heads} = 4$ | 1902 |
| THP | $h = 16, l = 2, \text{heads} = 4$ | 1756 |
| AttNHP | $h = 8, t = 2, l = 2, \text{heads} = 2$ | 1230 |
| IFTPP | $h = 16$ | 1965 |
| DLHP | $h = 8, p = 4, l = 2$ | 358 |

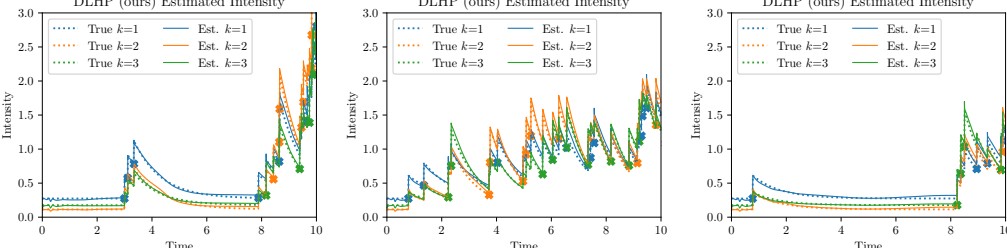

Figure 11: Our proposed DLHP model trained with 50k training sequences drawn from a randomly instantiated multivariate Hawkes process. Three example test sequences are plotted for each model.

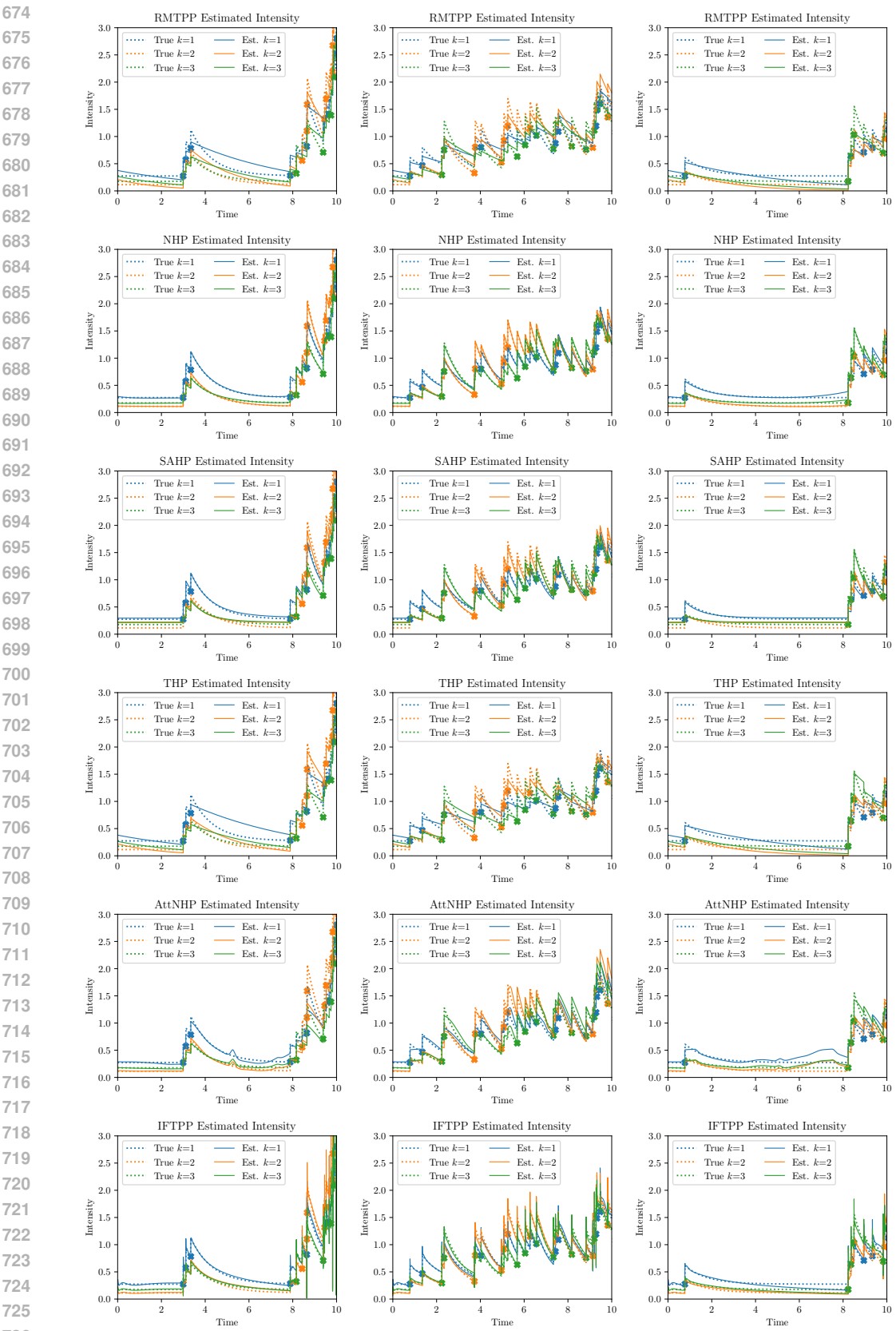

Figure 12: Baseline models trained with 50k training sequences drawn from a randomly instantiated multivariate Hawkes process. Three example test sequences are plotted for each model.

Table 14: [Added] Performance comparison of models on the synthetic Hawkes process experiment presented above. Higher log-likelihood indicates better performance, whereas lower root mean squared error (RMSE) indicates better performance.

| Model | Total Log-likelihood $\mathcal{L}_{\text{Total}}$ ($\uparrow$) | Next-Event Time RMSE ($\downarrow$) |
|---|---|---|
| RMTPP | -0.550 | 0.570 |
| NHP | -0.530 | 0.565 |
| SAHP | -0.537 | 0.569 |
| THP | -0.543 | 0.570 |
| AttNHP | -0.533 | 0.655 |
| IFTPP | -0.534 | 0.540 |
| DLHP (Ours) | -0.527 | 0.566 |

