# OpenReview forum: "Deep Linear Hawkes Processes"
_ICLR.cc/2025/Conference — Submitted to ICLR 2025_

### Official Review · Reviewer_AFMw · 2024-10-29

**Soundness:** 3
**Presentation:** 3
**Contribution:** 3
**Rating:** 6
**Confidence:** 4

**Summary:**

This paper introduces a novel marked temporal point process (MTPP) model called DLHP by drawing connections between linear Hawkes processes (LHPs) and deep state-space models (SSMs). Based on the SDE form of LHPs, DLHP models the intensity function by utilizing parameterization and parallelization techniques within SSMs. This approach achieves linear scalability and enables parallel processing, addressing the shortcomings of existing MTPP models. Experimental results demonstrate the effectiveness of DLHP.

**Strengths:**

1. Modeling the intensity function with the proposed latent linear Hawkes (LLH) layer is a novel and promising idea. The proposed model has the advantages of linear scalability and parallelism, addressing the shortcomings of existing marked TPPs.
2. The background and the proposed method are presented clearly. The paper is well-written and easy to follow.
3. Experiments are conducted to evaluate the model's predictive performance in terms of log-likelihood and computational efficiency.

**Weaknesses:**

1. While log-likelihood is an important metric for evaluating TPP model performance, this paper does not include experiments on the RMSE of time prediction—a metric commonly used in most baseline studies, such as RMTPP, SAHP, and THP.
2. Performing intensity recovery only on an inhomogeneous Poisson process is insufficient; it is also necessary to include recovery for other traditional TPPs, such as the Hawkes process.

Minor: The per-event log-likelihood in line 359 needs to be divided by the number of events N.

**Questions:**

1. If the background intensity is directly extended to be a function of time t, can the existence of a positive solution for Eq.(9) still be ensured?
2. Is it possible to use other numerical methods, such as the Euler method, to discretize the proposed LLH layer?

**Details Of Ethics Concerns:**

None.

---

> ### Author Response · Authors · 2024-11-19
> **Response to Reviewer AFMw**
>
> Thank you for taking the time to read and review our submission, and for your high soundness, presentation and contribution scores, and your positive comments about our method!
>
> Your questions were very on-point; so much so that other reviewers echoed all or parts of your questions! We therefore refer the reviewer to the Global Response for our initial response to comments that were shared between reviewers. We then provide more fine-grained responses here.
>
> > W1. While log-likelihood is…
>
> Please refer to the global response (“1. Log-Likelihood Results”).
>
> > W2. Performing intensity recovery…
>
> Please refer to the global response (“2. Synthetic Experiments”).
>
> > Minor: the per-event…
>
> Good spot, we have corrected this.
>
> > Q1. If the background…
>
> Yes, it can be through either constraining the background intensity function $\boldsymbol{\nu}_t$ or by rectifying the intensity by passing it into a non-negative activation function, similar to how we do in general for the DLHP in Eq. 13. You are correct, though, that in general, without either of these, it would not be possible to ensure a positive solution. **We have added clarification of this.**
>
> > Q2. Is it possible…
>
> Yes, it is possible to use other discretization methods. This flexibility was explored at length in the original S4 paper [Gu, Goel and Re, 2022, ICLR] and the follow-up work S4D [Gu, Gupta, Goel and Re, 2022, NeurIPS]. It is our understanding that there is comparatively little systematic difference in the performance between the discretization methods. **We have added a note of this.**
>
> Thank you again for taking the time to read and review our submission. Please do not hesitate to reach out if you have further questions!
>
> -- Submission 3981 Authors

---

> > ### Comment · Reviewer_AFMw · 2024-11-26
> >
> > Thank you for your response and the additional experiments. The reply addresses my concerns, and I have raised my score to 6.

---

> > > ### Author Response · Authors · 2024-11-28
> > > **Response to Reviewer AFMw**
> > >
> > > Dear Reviewer AFMw,
> > >
> > > Thank you for your feedback and for raising your score. We've addressed all your comments and have uploaded a fully revised manuscript for your reference.
> > >
> > > If you have any further questions or concerns that would prevent you from enthusiastically supporting our paper, please don't hesitate to ask before the discussion window closes. If not, given your positive feedback and the addressed concerns, we would be grateful if you would consider raising your score further.
> > >
> > > Thank you again for your time and helpful feedback.
> > >
> > > -- Submission 3981 Authors

---

### Official Review · Reviewer_15be · 2024-10-30

**Soundness:** 3
**Presentation:** 3
**Contribution:** 2
**Rating:** 5
**Confidence:** 3

**Summary:**

This research paper introduces the Deep Linear Hawkes Process (DLHP), a novel approach to modeling Marked Temporal Point Processes (MTPPs) that bridges the gap between deep state-space models and MTPPs. The benefits of state space models (SSMs) compared to attention-based model were emphasized. The innovation lies in modifying stochastic jump differential equations that describes the intensity function $\lambda$ of MTPP to become linear differential equations in deep state-space models. Its applicability was tested on various benchmark dataset. Notably, this work marks the first successful adaptation of state-space model architectural features to create a new category of MTPP models.

**Strengths:**

The paper introduces a new family of MTPP models that leverages SSM layers. Also, the stochastic jump differential equation was translated in terms of SSM. By stacking multiple layers more flexible intensity function can be generated, while input dependent characteristic enabled more complex behavior.

**Weaknesses:**

Only NLL was used as an evaluation metric, whereas RMSE and accuracy is commonly used in prior works. Also, as a method for modeling a stochastic process, simulation study can help showing the effectiveness of the method.

**Questions:**

- In Fig. 4, THP and IFTPP shows faster runtime when implemented in JAX. Is DLHP still faster when they are all implemented using JAX?
- Demonstration of the simulation study might be helpful to see wether the discretization of DLHP does not hurt the expressivity.

---

> ### Author Response · Authors · 2024-11-19
> **Response to Reviewer 15be**
>
> We again thank the reviewer for taking the time to read and review our paper!  We refer the reviewer to the Global Response for our initial response to comments that were shared between reviewers. We then provide more fine-grained responses here.
>
> > Only NLL was…
>
> Please refer to the global response. We do stress, however, that we presented numerous metrics: the total log-likelihood, both time and mark log-likelihoods, classification accuracy for next event type prediction, and time and mark classification calibration scores. These metrics were presented in full in the appendix. **We will add text to the main body highlighting and summarizing these additional results.**
>
> > Also, as a…
>
> Can the reviewer please clarify what they mean by a “simulation study” in this context? If you are referring to fitting the model to a synthetic dataset with a known ground truth intensity, then we are currently evaluating this (see the global response). If you are referring to something different, then please let us know and we will try and include it!
>
> > Q1. In Fig. 4, THP…
>
> THP and IFTPP were not implemented in JAX, instead using the EasyTPP PyTorch implementation. The EasyTPP IFTPP implementation uses the highly optimized, low-level GRU implementation in PyTorch, and so would not benefit from implementing in JAX.  We suspect THP may improve in runtime with JAX.  However, the main drawback with THP is memory and work scaling; both of which are present in any implementation of the model and are both seen in Figure 4.
>
> At the time of writing, PyTorch did not ship an easy-to-use complex-valued parallel scan method. We therefore implemented a standalone DLHP architecture in JAX to allow us to fairly test the performance with the inbuilt complex-valued parallel scan in JAX.
>
> Since submission, we have actually implemented a pure PyTorch and Triton backend for the parallel recurrence for a real-valued DLHP (since Triton does not support complex values yet). We find that our Triton-PyTorch hybrid implementation is faster than the EasyTPP baselines at longer sequence lengths. We do note, however, our Triton implementation is slower at shorter sequence lengths. We think this is to do with an inefficiency in the compilation/deployment of our Triton kernel. We are actively working to improve this now. Even still, **this highlights that DLHP admits faster inference on long sequences in the same environment.**
>
> > Q2. Demonstration of the…
>
> Can the reviewer please clarify what they would like us to explore in this comment?
>
> If you are asking if there is an accuracy drop through discretization, then the answer is that there is no drop in accuracy. The model is instantiated and learned using the chosen discretization strategy (i.e. discretized at event times) and hence learns representations that eliminate inaccuracies (as opposed to discretizing an existing model, which may incur approximation error and a subsequent performance drop). If the reviewer is asking us to benchmark different discretization strategies, then this is something we could certainly look into. However, we note that most SSM papers do not find systematic performance differences with different discretization strategies [Gu, Goel and Re, 2022, ICLR;  Gu, Gupta, Goel and Re, 2022, NeurIPS].
>
>
> Thank you again for taking the time to read and review our submission. Please do not hesitate to reach out if you have further questions!
>
> -- Submission 3981 Authors

---

> > ### Comment · Reviewer_15be · 2024-11-25
> >
> > Thank you for your response to my review and for conducting additional experiments to simulate the Hawkes process with various parameters. However, I believe your statement that DLHP and NHP are the only methods capable of recovering the ground truth intensity requires further clarification. For example, providing metrics such as the mean squared error (MSE) between the ground truth and the simulated process could strengthen your argument and offer more concrete evidence.

---

> ### Author Response · Authors · 2024-11-25
> **Response to Reviewer 15be**
>
> Thank you for your reply! You are right, on reflection, our wording was a little bit strong and this was not our intention. What we intended was that both NHP and DLHP better recover the ground truth intensities when visually inspecting the results (compared to other baselines).  We included these plots (with the ground truth intensities) at the end of the **Appendix (page 31-32)**.  That said, it is definitely true that all models are doing a decent job of recovering the ground truth intensities (just that NHP and DLHP are slightly better visually). We have updated our initial response to reflect your suggestions.
>
> It is a great idea to also assess the synthetic application quantitatively. We have computed MSE and per-event log-likelihood on a held-out test set. Please find both results in the table below. We find that DLHP is again competitive on both metrics.
>
> We really appreciate your input, which has definitely strengthened our paper and further highlighted the performance of our DLHP model!
>
>
> | **Model** | MSE  ↓  | Log-likelihood  ↑  |
> |-|-|-|
> | **RMTPP** | 0.325 | -0.550 |
> | **NHP** | 0.319| -0.530 |
> | **SAHP** | 0.324 | -0.537 |
> | **THP** | 0.325 | -0.543 |
> | **AttNHP** | 0.429 | -0.533 |
> | **IFTPP** | 0.292 | -0.534|
> | **DLHP (Ours)** | 0.320 | -0.527 |

---

> ### Author Response · Authors · 2024-11-28
> **Response to Reviewer 15be**
>
> Dear Reviewer 15be,
>
> Thank you for your engagement with the updated material. With the discussion period closing soon, we're wondering if you happened to have a chance to review the additional experiments you suggested. We've addressed all your comments and have uploaded a fully revised manuscript for your reference. If you have any further questions, please don't hesitate to ask.
>
> If you're satisfied with the additional experiments and have no further questions, we would be grateful if you would consider raising your score.
>
> Thank you again for your time and helpful feedback.
>
> -- Submission 3981 Authors

---

### Official Review · Reviewer_CrNn · 2024-11-04

**Soundness:** 3
**Presentation:** 3
**Contribution:** 2
**Rating:** 6
**Confidence:** 5

**Summary:**

This paper presents an interesting idea of using deep state-space models (SSMs) to model marked temporal point processes. The authors draw a connection between the SSMs and linear Hawkes processes via their stochastic differential equations, and develop a tailored model that enjoys comparable or superior performance against neural marked TPP baselines. The overall presentation is clear and easy to follow.

**Strengths:**

The idea of combining the deep SSMs with marked temporal point processes is interesting. The methodology/model architecture is well presented, easy for the readers to understand how these models are integrated together, via techniques like diagoalization, discretization, input-dependent parameters, etc. Meanwhile, the new model enjoys a better computational efficiency than other baselines when implemented with parallel scan.

**Weaknesses:**

Though I enjoyed reading the paper, there are still a few aspects of the paper that I think can be improved or answered by the authors:

*Methodology*:

1. I am not very convinced about the use of the Hawkes-process-like mechanism in each layer. I understand the connection between a linear Hawkes process and the SSM via the stochastic differential equation, but what is the need/meaning/interpretation of using this connection in the model? Why use $AB$ instead of one matrix of size $P\times H$ (or use $E\alpha$ instead of one $P*K$ matrix) as the parameters in Equation 10? If the idea is to inherit or imitate the parameter structure in the linear Hawkes process, this structure of parameters is overwritten/ignored since there are multiple layers and non-linear transformations stacked together.

2. On the other hand, if the authors do want to imitate the parameter structure of the linear Hawkes process, then the diagonalization would assume no influence between different marks, which is unrealistic in practice. And there is also no interpretation of the learned $\Lambda$ and $\alpha$ in the experiments.

*Experiments*:

3. I would suggest that the authors conduct more synthetic data experiments and results to validate the model’s ability to recover the ground truth. Only one simple Poisson process example is not enough, and the real data experiments are not straightforward in terms of interpreting the model's effectiveness.

4. No standard deviation was reported for the metrics in the tables/figures. The results of DLHP look like a random win against baselines given no reported standard deviation, especially only on real-world datasets with no ground truth.

5. It would be better if there was a rigorous computational complexity analysis of the model and baselines since the authors claim that it is one of the main strengths of using SSMs in modeling marked TPPs.

**Questions:**

1. The main text says $u_t^{(1)} = 0$ (line 231), but in figure 2 it says $u_t^{(1)}$ is the event embedding. I am confused about this.

2. Do the authors have any insights on how to choose the model architecture, given the performance discrepancy of the model using forward and backward mechanisms?

3. What do the authors mean by 1.4 times improvement from baselines in line 92? If it refers to the average rank of the DLHP, does it mean a lower number is better?

---

> ### Author Response · Authors · 2024-11-19
> **Response to Reviewer CrNn (1/2)**
>
> Thank you for taking the time to provide a number of really insightful observations and questions! We were very glad to hear you enjoyed reading our submission!
>
> We refer the reviewer to the Global Response for our initial response to comments that were shared between reviewers. We then provide more fine-grained responses here.
>
> > W1. I am not very convinced …
>
> Thank you for these great questions. We started this work by trying to incorporate deep SSMs into MTPPs.  Fully leveraging the continuous-time dynamics of deep SSMs — as is so natural in an MTPP context to handle variable inter-arrival times — _necessitated_ using discrete impulses and converting the ODE into a stochastic jump process. The resulting process bore a strong resemblance to the canonical LHP.  This provided a lens to situate our new model in the existing literature by matching the parameterization wherever possible.  ⁠Crucially, these parameterizations were not “design decisions”, but rather natural touchpoints to mathematically define how our model is similar and where it differs.
>
> You are then absolutely right that certain matrices can be factorized in several different ways; but this possibility does not materially affect the theoretical complexity, implementation, etc. of the model. This was a byproduct of drawing connections to the LHP. Indeed, any such factorization could be used – we do not claim this factorization is “better” for performance. For instance, $\mathbf{ABu}_t dt$ resulted from mapping $\boldsymbol{\beta}$ to $\mathbf{A}$ and $\boldsymbol{\nu}_t$ to $\mathbf{Bu}_t$ from the original term $\boldsymbol{\beta}\boldsymbol{\nu}_t dt$ term in the LHP.
>
> We did however introduce $\mathbf{E} \boldsymbol{\alpha}$ to facilitate a low-rank factorization of the mark embeddings (i.e. $\mathbf{E}$ is a tall, narrow matrix). If there is a large mark space (as in EHRShot) then the parameter count in the embedding matrix would be enormous.
>
> > W2. On the other hand, ...
>
> We believe this is a little bit of a misunderstanding. In the core formulation in Equation (10), the latent dimensions can interact, as $\mathbf{A}$ is dense. We _then_ diagonalize the system through a linear rotation into (and out of) the eigenbasis. This diagonalization _does not_ modify the underlying dynamics.
>
> Put another way: while each dimension of $\mathbf{x}$ is independent in time after diagonalization, the channels are mixed through the input matrix $\mathbf{B}$, the emission matrix $\mathbf{C}$, and the position-wise nonlinearity. Furthermore, events with a specific mark $m$ affect all dimensions of the hidden state through the mark-specific impulse $\mathbf{E}\boldsymbol{\alpha}_m$.Therefore, although each latent dimension is independent (i.e. they are in the eigenbasis of the layer), the marks actually do communicate because the eigenbasis of different layers is different, because of the mixing in the projection matrices, and because of the position-wise non-linearities.
>
> Marks would only be independent if every DLHP layer, in the notation of Equation (10), were restricted to have: a diagonal input matrix $\mathbf{B}$, diagonal output matrix $\mathbf{C}$, diagonal feedthrough matrix $\mathbf{D}$, and diagonal effective impulse matrix $\mathbf{E}\boldsymbol{\alpha}$ for all layers (as well as a diagonal $\mathbf{W}$ to transform the last layer’s output into intensity values). To be clear, this would be a _very_ severe set of additional constraints on our proposed model.
>
> As for interpretability: you are correct that $\boldsymbol\Lambda$ and $\boldsymbol\alpha$ have limited interpretability in and of themselves. That being said, we have included a more general discussion of interpretability in the response to Reviewer PxVB (search for _“W2. The proposed method”_).
>
> **We have clarified all of these points at the end of the methods section**. We are more than happy to answer any more questions you have on this topic!
>
> > W3. I would suggest…
>
> Please refer to the global response.
>
> > W4. No standard deviation…
>
> Please see the global response. The key takeaway is that DLHP still performs comparably or better than the baseline models across multiple random seeds. Thank you for highlighting this.

---

> ### Author Response · Authors · 2024-11-19
> **Response to Reviewer CrNn (2/2)**
>
> # (Continued)
>
> > W5. It would be better…
>
> This is a great suggestion.  We have included a table below (and in the manuscript) defining the work, memory and parallelism of the methods in Big-O notation. $L$ denotes the sequence length, and $M$ denotes the number of Monte Carlo samples used in evaluating $\int \lambda_t d t$ in Eq. (2). As IFTPP is an intensity-free method with parametric assumptions, it does not need to estimate $\int \lambda_t d t$ as the other methods do.
>
> **The crucial point is that DLHP matches the best-performing baseline in every category.** The empirical speed evaluations presented in Figure 4 of the original submission corroborate these results.
>
> | Method | Forward Pass | Forward Pass | Forward Pass | Estimating $\int \lambda_t d t$| Estimating $\int \lambda_t d t$ | Estimating $\int \lambda_t d t$| Overall |
> |-|-|-|-|-|-|-|-|
> | | Memory | Work | Parallelism | Memory | Work | Parallelism | Parallelism |
> | RMTPP | $\mathcal{O}(L)$ | $\mathcal{O}(L)$ | $\mathcal{O}(L)$ | $\mathcal{O}(LM)$ | $\mathcal{O}(LM)$ | $\mathcal{O}(1)$ | $\mathcal{O}(L)$ |
> | NHP | $\mathcal{O}(L)$ | $\mathcal{O}(L)$ | $\mathcal{O}(L)$ | $\mathcal{O}(LM)$ | $\mathcal{O}(LM)$ | $\mathcal{O}(1)$ | $\mathcal{O}(L)$ |
> | SAHP | $\mathcal{O}(L^2)$ | $\mathcal{O}(L^2)$ | $\mathcal{O}(\log L)$ | $\mathcal{O}(LM)$ | $\mathcal{O}(LM)$ | $\mathcal{O}(1)$ | $\mathcal{O}(\log L)$ |
> | THP | $\mathcal{O}(L^2)$ | $\mathcal{O}(L^2)$ | $\mathcal{O}(\log L)$ | $\mathcal{O}(LM)$ | $\mathcal{O}(LM)$ | $\mathcal{O}(1)$ | $\mathcal{O}(\log L)$ |
> | AttNHP | $\mathcal{O}(L^2)$ | $\mathcal{O}(L^2)$ | $\mathcal{O}(\log L)$ | $\mathcal{O}(L^2M)$ | $\mathcal{O}(L^2M)$ | $\mathcal{O}(\log L)$ | $\mathcal{O}(\log L)$ |
> | IFTPP | $\mathcal{O}(L)$ | $\mathcal{O}(L)$ | $\mathcal{O}(L)$ | N/A | N/A | N/A | $\mathcal{O}(L)$ |
> | DLHP | $\mathcal{O}(L)$ | $\mathcal{O}(L)$ | $\mathcal{O}(\log L)$ | $\mathcal{O}(LM)$ | $\mathcal{O}(LM)$ | $\mathcal{O}(1)$ | $\mathcal{O}(\log L)$ |
>
> > Q1. The main text…
>
> Thank you for highlighting this. This is a slight oversight in Figure 2. The input, $\mathbf{u}_t$, is $\mathbf{0}$ for layer one (the input solely comes through the mark impulses $\mathbf{E}\boldsymbol{\alpha}d\mathbf{N}_t$ on layer one). We used $\mathbf{u}$ to indicate inputs and events here, but in hindsight, this is not clear. **We have updated the caption to explicitly state this**.
>
> > Q2. Do the authors…
>
> At the time of writing, we do not have insight on this. We explored multiple avenues, e.g. is one preferable or larger/smaller mark spaces, heavier-tailed inter-arrival time distributions, etc, but found no compelling insights. We then treated it as a hyperparameter search (see Table 9), and then selected backwards discretization (as it was slightly better on average) to use as a single, unified architecture for our main metrics table. **We have added exploring this as an avenue for future work in the conclusion.**
>
> > Q3. What do the…
>
> The 1.4 corresponds to the average multiplicative improvement in the likelihood between our method and the best baseline, so higher is better. For the tables that we produce, higher log-likelihood values are better, whereas a lower rank is better (with rank = 1 being the best). **We have clarified both in the text.**
>
> Thank you again for taking the time to read and review our submission. We hope our comments have allayed your concerns. Please do not hesitate to reach out if you have further questions!
>
> -- Submission 3981 Authors

---

> > ### Comment · Reviewer_CrNn · 2024-11-27
> >
> > I appreciate the authors response to my review and I can see the efforts that the authors have put in. I think all my questions have been properly answered.
> >
> > However, with all the responses being added during the rebuttal period, I think there is a large portion of effort that required for the authors to appropriately incorporate them into the paper and re-think how to organize and revise the content. I am not quite convinced that the current version has been fully ready for publication in ICLR.
> >
> > To this end, I decide to raise my score to marginally above the threshold, but will leave the final decision to the ACs/SACs. I wish the authors good luck while they keep improving this study.

---

> > > ### Author Response · Authors · 2024-11-28
> > > **Response to Reviewer CrNn**
> > >
> > > Dear Reviewer CrNn,
> > >
> > > Thank you for your feedback and positive leaning towards acceptance. We've addressed all your comments and have uploaded a revised PDF for your reference.
> > >
> > > Regarding your concern about the scope of the additions, we want to clarify that these additions strengthen and clarify the existing content without requiring substantial reorganization. All changes are either additional supplementary results tables in the Appendix or explicit clarifications to address any misunderstandings or ambiguities. The revised manuscript, with changes highlighted in blue, adheres to the 10-page limit.
> > >
> > > We would love for you to be enthusiastic about the acceptance of our paper in light of this fully revised version and your otherwise positive review.
> > >
> > > Irrespective of this, thank you very much for taking the time to review our submission, and for providing both positive and constructive feedback!
> > >
> > > Thank you again for your time and helpful feedback.
> > >
> > > -- Submission 3981 Authors

---

### Official Review · Reviewer_PxVB · 2024-11-04

**Soundness:** 3
**Presentation:** 3
**Contribution:** 3
**Rating:** 6
**Confidence:** 3

**Summary:**

This paper introduces a new marked temporal point process (MTPP) model known as the deep linear Hawkes process (DLHP), which is developed by integrating deep state-space models (SSMs) and linear Hawkes processes (LHPs). Through the use of diagonalization and discretization techniques, the DLHP enables efficient implementation of closed-form update. The model has been shown to surpass current methods in terms of performance on a variety of benchmark tasks, excelling in metrics such as log-likelihood and runtime across different sequence lengths. A key contribution of the research is pioneering the use of SSMs' unique architecture to develop a new class of MTPP models, positioning the DLHP as a competitive model in applications.

**Strengths:**

1. This work stands out for being the first effort in harnessing the distinctive architectural features of deep state-space models (SSMs) to develop marked temporal point process (MTPP) models.
2. Efficient Implementation: the DLHP has demonstrated robustness, performance, computational efficiency, and extensibility in experiments. These qualities seem to make the DLHP a competitive contender as an out-of-the-box model for a variety of applications.
3. Clear presentation: This paper presents its idea clearly.

**Weaknesses:**

1. As the authors clarified, DLHP has a higher implementation complexity, and the model architectures for DLHP tends to be more complex compared to other models. Does the enhancement in performance contributes to this increased complexity (since increased model complexity may lead to higher model likelihood value)? It would be better to provide necessary analysis regarding this issue and add some experiments for illustration if possible.
2. The proposed method sacrifices some of the interpretability of latent dynamics and parameters that are characteristic of LHPs, which could impact the model's transparency and understanding, and limit its usability for high-stakes applications where understanding event interactions is crucial. Would you discuss possible solutions or ideas regarding this issue? The motivation stated in line 201-214 of page 4 may provide some clues and could be discussed more deeply.

**Questions:**

The authors may also pay attention to the paper titled with “Mamba Hawkes Process”（https://arxiv.org/abs/2407.05302）and do a detailed comparison with it, also to further confirm the statement "the first instance of the unique architectural capabilities of SSMs being leveraged to construct a new class of MTPP models" in the paper.

See the weaknesses for other questions.

**Details Of Ethics Concerns:**

-

---

> ### Author Response · Authors · 2024-11-19
> **Response to Reviewer PxVB**
>
> Firstly, we thank the reviewer for their positive comments towards our work, particularly highlighting how we are the first to fully leverage the opportunities of deep SSMs for TPPs and the presentation of our work.
>
> We refer the reviewer to the Global Response for our initial response to comments that were shared between reviewers. We then provide more fine-grained responses here.
>
> > W1. As the authors clarified ...
>
> We apologize, there might have been a lack of precision in our language for our submission. In line 480, we use the complexity of the implementation to refer to the fact that the fast implementation requires a parallel scan implementation, as well as a conventional sequential scan, i.e. a for-loop-based implementation. This was in contrast to other models, such as the neural Hawkes process (NHP), which only supports a sequential scan. This is purely an implementation detail, and doesn’t change the expressivity of the model family or parameter counts of the model. **We have refined the language throughout the paper when discussing these points.** Thank you for highlighting this oversight.
>
> It is worth also noting that throughout our experiments we have controlled the parameter count of different models to ensure roughly equivalent model capacity across methods and datasets. If neither of these points addresses your original concerns please let us know and we can try to clarify!
>
>
> > W2. The proposed method …
>
> Thank you for this question, it is something we actually thought about prior to submission. Similar to other neural MTPP methods, interpreting the model parameters, latent states etc is extremely difficult. However, the unique SSM formulation does offer some limited opportunities for interpretation: due to the LLH layers effectively writing to and from a residual stream with mark-dependent impulses, we can build a coarse approximation of the instantaneous interaction of mark types by summing the impulses across the layers analogous to the original interpretation of the $\boldsymbol{\alpha}$ parameters of linear Hawkes processes. However, doing so requires neglecting certain non-linearities and temporal dependencies, and hence is a very crass approximation.
>
> However, your question on interpretability has an opportunity beyond just interpreting the parameters:
> - As the DLHP is stateful, we can approach interpretability by dimensionality reduction and clustering/decoding of the latent states (see e.g. Mitchell-Heggs et al [2022, J Comput Neurosci] or Maheswaranathan et al [2019, NeurIPS]). This type of analysis is not afforded by attention-based methods. These methods provide richer descriptions and analyses of trajectories and model computation.
> - As the LLH is a deep SSM, as interpretability and understanding of deep SSMs grows, our DLHP automatically inherits these interpretability methods.
>
> **We will include a discussion of these points in any camera-ready version of the paper.**
>
> > Q1. The authors may …
>
> We did briefly discuss the contemporaneous (under ICLR’s definition [here](https://iclr.cc/Conferences/2025/FAQ)) Mamba Hawkes Process (MHP) in the related work section. To clarify our statement, we were specifically referencing how our model both encodes events and decodes intensity values with the _same SSM_, whereas the MHP uses Mamba only as an event encoder. MHP intermediate intensities are decoded as a fixed parametric function of time and the hidden state. Contrasting this with our model, the same latent dynamics are used to both encode events through impulses and evaluate intermediate intensities through continuous-time differential equations. This is why we describe the DLHP as being the first method to *fully* leverage the capabilities of deep SSMs, as opposed to using a deep SSM as a highly performant encoder. **We have expanded this discussion in the main paper.**
>
> Thank you again for taking the time to read and review our submission. Please do not hesitate to reach out if you have further questions!
>
> -- Submission 3981 Authors

---

> > ### Comment · Reviewer_PxVB · 2024-11-27
> >
> > Thanks for your response and the additional analysis regarding the interpretability. The reply addresses my concern about the complexity and the whole rebuttal makes the claim of this paper clearer. With overall consideration, I will maintain my score of 6.

---

> > > ### Author Response · Authors · 2024-11-28
> > > **Response to Reviewer PxVB**
> > >
> > > Dear Reviewer PxVB,
> > >
> > > Thank you for your valuable feedback and positive vote towards acceptance. We've carefully addressed all your comments and have uploaded a revised PDF for your reference.
> > > We kindly ask if you have any remaining concerns that would prevent you from giving a stronger vote for acceptance. If not, given your positive feedback and high scores, we would be grateful if you would consider raising your overall rating.
> > > Thank you again for your time and insightful review.
> > >
> > > – Submission 3981 Authors

---

### Author Response · Authors · 2024-11-19
**Global Response**

# Global Response

Foremost, we thank all four reviewers for taking the time to thoroughly read and provide detailed feedback for our submission.

We presented the Deep Linear Hawkes Process (DLHP), a novel adaptation of deep state-space models (SSMs) to marked temporal point process (MTPP) applications. Our model draws connections between classical Hawkes processes and deep SSMs, allowing us to use the unique capabilities of deep SSMs to parsimoniously define a point process model with linear work, parallelization, and without the need for additional decoding heads. **We find our model reliably matches or outperforms other methods in terms of: log-likelihood (mark, time, and total), runtime, next mark prediction accuracy, next mark calibration, and inter-arrival time calibration.**

We were very heartened to see all four reviewers praise and score highly the soundness and presentation of our work. The reviewers all then commented on the various positives of our method. We are very happy that all four reviewers clearly engaged deeply with our work, and feel they accurately captured the core contributions of our submission.

We first comment on any feedback that was shared between reviews below, and directly respond to individual reviewer feedback and questions in separate comments. We are still working on some additional experiments and evaluations, as requested by the reviewers. We will post updates as a further comment when these are complete. These results, as well as promised revisions to the text, will be incorporated in an updated manuscript that we will upload before the end of the rebuttal period.

## 1. Log-Likelihood Results (Table 1) with Repeats

Reviewer CrNn requested repeatability results for our main set of experiments to account for uncertainty. We re-conducted our main log-likelihood experiments (Table 1) across five different random seeds and computed the mean (and standard deviation) for each model and dataset. We present these results here in a global response as we feel that these may be of interest to all reviewers. Higher ↑ log-likelihoods are better; lower ↓ average ranks are better. The best method is bolded, and the second-best method is shown in italics. **The key takeaway is that DLHP is still the best or second-best performing method, evidenced by achieving the best (lowest) average rank.**

| **Model** | **Amazon** ↑ | **Retweet** ↑ | **Taxi** ↑ | **Taobao** ↑ | **StackOverflow** ↑ | **Last.fm** ↑ | **MIMIC-II** ↑ | **EHRShot** ↑| **Avg. Ranking** ↓ |
|-|-|-|-|-|-|-|-|-|-|
| **RMTPP** | -2.136 (0.003) | -7.098 (0.217) | 0.346 (0.002) | 1.003 (0.004) | -2.480 (0.019) | -1.780 (0.005) | -0.472 (0.026) | -8.081 (0.025) |  6.1  |
| **NHP** | 0.129 (0.012) | **-6.348** (0.000) | _0.514_ (0.004) | 1.157 (0.004) | -2.241 (0.002) | -0.574 (0.011) | 0.060 (0.017) | _-3.966_ (0.058) | _2.9_  |
| **SAHP** | -2.074 (0.029) | -6.708 (0.029) | 0.298 (0.057) | -1.64 (0.083) | -2.341 (0.058) | -1.646 (0.083) | -0.677 (0.072) | -6.804 (0.126) |  5.6 |
| **THP** | -2.096 (0.002) | -6.659 (0.007) | 0.372 (0.002) | -1.712 (0.011) | -2.338 (0.014) | -1.712 (0.011) | -0.577 (0.011) | -7.208 (0.096) | 5.5  |
| **AttNHP** | 0.484 (0.077) | -6.499 (0.028) | 0.493 (0.009) | 1.259 (0.022) | -2.194 (0.016) | -0.592 (0.051) | -0.170 (0.077) | OOM | 3.1  |
| **IFTPP** | _0.496_ (0.002) | -10.344 (0.016) | 0.453 (0.002) | **1.318** (0.017) | _-2.233_ (0.009) | **-0.492** (0.017) | _0.317_ (0.052) | -6.596 (0.240) | _2.9_  |
| **DLHP (Ours)** | **0.781** (0.011) | _-6.365_ (0.003) | **0.522** (0.004) | _1.304_ (0.039) | **-2.163** (0.009) | _-0.557_ (0.046) | **1.243** (0.083) | **-2.512** (0.369) | **1.4** |


## 2. Synthetic Experiments

Reviewers CrNn and AFMw raise the good point of applying DLHP to synthetic datasets.  **We are currently in the process of benchmarking the performance of the DLHP against the true model and baselines on synthetic datasets drawn from randomly instantiated parametric temporal point processes.** We will report back on these results before the end of the rebuttal period, and commit to including them in any camera-ready paper.


## 3. RMSE

Reviewers 15be and AFMw also requested the evaluation of the RMSE metric from our experiments. **We are currently evaluating the RMSE for all the methods, and will report back on the results before the end of the rebuttal period.** We commit to including the RMSE results in any potential camera-ready paper.


## Summary

We again thank the reviewers for taking time to read and review our paper. We will add another comment shortly when the remaining experimental results come in. We hope these comments and clarification help allay any questions you have. Please do not hesitate to reach out if you have further questions!

-- Submission 3981 Authors

---

### Author Response · Authors · 2024-11-25
**Additional Experimental Results**

# Follow-Up for Global Response


## 1. Log-Likelihood Results (Table 1) with Repeats
We have now updated the full log-likelihood results. **Our DLHP model is consistently comparable with or outperforms other baselines across all datasets.**


## 2. Synthetic Experiments
As Reviewers CrNn and AFMw suggested, we have conducted additional experiments on multivariate Hawkes processes where the ground truth intensities are available. We have included the **full plots and details at the end of Appendix (page 31-32) in the revision**. After the discussion period, we will move it to the next subsection of synthetic Poisson experiments. **The main takeaway is that our model is able to recover the ground truth almost perfectly with the fewest parameters.** All models are doing a decent job recovering the ground truth intensities, where NHP and DLHP visually better recover the ground truth intensities.


## 3. RMSE
Reviewers 15be and AFMw suggested including results for RMSE; we have performed the RMSE evaluation on all methods (lower is better, OOM refers to out-of-memory):

| **Model** | **Amazon**  | **Retweet**  | **Taxi**  | **Taobao**  | **StackOverflow**  | **Last.fm**  | **MIMIC-II**  | **EHRShot** |
|-|-|-|-|-|-|-|-|-|
| **RMTPP** | 0.361 | 16152 | 0.284 | 0.127 | 1.054 | 15.864 | 0.754 | 3445 |
| **NHP** | 0.342 |  15322 | 0.283 | 0.127 | 1.028 | 15.841 | 0.739 | 3430 |
| **SAHP** | 0.345 |  16018 | 0.285 | 0.126 | 1.034 | 15.830 | 0.805 | 3418 |
| **THP** | 0.335 |  15848 | 0.286 | 0.126 | 1.029 | 15.878 | 0.781 | 3504 |
| **AttNHP** | 1.214 |  16220 | 1.273 | 0.130 | 1.311 | 15.889 | 0.852 | OOM |
| **IFTPP** | 0.332 |  2568 | 0.280 | 0.126 | 0.975 | 15.550 | 0.713 | 1899 |
| **DLHP (Ours, sampling)** | 0.395 | 15223 | 0.282 | 0.127 | 1.021 | 15.844 | 0.764 | 3421 |
| **DLHP (Ours, linear probe)** | 0.337 | 14642 | 0.283 | 0.128 | 1.038 | 15.710 | 0.810 | 3350 |

For all baseline methods, results were computed with empirical estimates of expected values via Monte Carlo samples; intensity-based methods were sampled using the thinning algorithm [1]. For our method, we showcase both sampling-based results as well as using a simple linear probe on the output hidden vector to directly estimate the next event time.
In general, we find that our method is comparable to the other intensity-based methods, and do not find substantial differences in RMSE between sampling and the linear probe. That being said, the linear probe was substantially faster (often around 10 times quicker) even when accounting for fine-tuning the linear probe parameters.

We would like to note, for context, that we did not originally include RMSE results for next-event time as it purely describes how well the model captures the expected value of the next-event time [2]. This is not informative when the inter-event times have a multimodal distribution—which we often encounter in practice with real data. Conversely, the time log-likelihood is a proper scoring rule [2,3] and handles multimodal distributions well. As we note in our paper, Daley & Vere-Jones [4] state “testing the model on the basis of its _forecasting performance_ amounts to testing the model on the basis of its _likelihood_” (emphasis added). Following the recommendations from Bosser, T. and Taieb, S.B. [5], we also report the total, time, and mark log-likelihoods, the time and mark calibration, as well as the next-event mark accuracy (see Appendix of original submission) for a comprehensive evaluation of the predictive/forecasting performance.

[1] Ogata, Yosihiko. “On Lewis' simulation method for point processes.” _IEEE transactions on information theory 27_(1)  (1981): 23-31.

[2] Gneiting, Tilmann. “Making and evaluating point forecasts.” _Journal of the American Statistical Association_ 106.494 (2011): 746-762.

[3] Gneiting, Tilmann, and Matthias Katzfuss. “Probabilistic forecasting.” _Annual Review of Statistics and Its Application 1_(1) (2014): 125-151.

[4] Daley, Daryl J., and David Vere-Jones. _An introduction to the theory of point processes: volume I: elementary theory and methods_ (page 276). Springer New York, 2003.

[5] Bosser, T. and Taieb, S.B., “On the Predictive Accuracy of Neural Temporal Point Process Models for Continuous-time Event Data.” _Transactions on Machine Learning Research_, 2023.


## Summary
We thank again for the detailed feedback by the reviewers. We will incorporate all updates and new experimental results into the revised version of this paper. Please feel free to let us know if these answer your questions or if you have further questions!


-- Submission 3981 Authors

---

### Meta-Review · Area_Chair_hBDm · 2024-12-23

**Metareview:**

This paper introduces the Deep Linear Hawkes Process (DLHP), combining deep state-space models (SSMs) with linear Hawkes processes for marked temporal point process modeling. The authors claim DLHP offers linear scaling and parallelism advantages over existing models, reporting competitive or superior performance across various metrics on real-world datasets.

The paper's strengths include the integration of deep SSMs with MTPPs and clear methodology presentation. Reviewers noted the competitive performance on benchmark datasets and potential for improved computational efficiency.

Weaknesses identified include limited interpretability compared to classical Hawkes processes and insufficient synthetic experiments. Initially, there was a lack of comprehensive evaluation metrics beyond log-likelihood, which was later addressed in the rebuttal.

The primary reason for rejection is that the paper, while presenting an interesting combination of existing techniques, does not demonstrate sufficient novelty or practical impact to warrant publication at ICLR. As noted by the Area Chair, the work appears to be a straightforward combination of SSMs and stochastic jump differential equations, which is somewhat incremental. While the model shows competitive performance, the improvements are marginal and do not clearly justify the increased complexity.

**Additional Comments On Reviewer Discussion:**

During the rebuttal period, reviewers requested additional experimental results and clarifications on model interpretability. The authors responded with additional log-likelihood results, synthetic experiments, RMSE evaluation, and clarifications on model architecture.

While the authors' thorough responses addressed many concerns, these additions did not fundamentally change the paper's core contribution or address the main concern about incrementality raised by the Area Chair. The additional experimental results strengthened the empirical validation but did not significantly elevate the paper's overall contribution.

Ultimately, despite the authors' efforts to address reviewer concerns, the fundamental issue of limited novelty and practical significance remained, aligning with the Area Chair's initial assessment of the work as an incremental combination of existing techniques.

---

### Decision · Program_Chairs · 2025-01-22

Reject